# Debiasing Averaged Stochastic Gradient Descent to handle missing values

**Aude Sportisse**
Sorbonne University
Paris, France
`aude.sportisse`
`@sorbonne-universite.fr`

**Claire Boyer**
Sorbonne University
Paris, France
`claire.boyer`
`@sorbonne-universite.fr`

**Aymeric Dieuleveut**
Ecole Polytechnique
Palaiseau, France
`aymeric.dieuleveut`
`@polytechnique.edu`

**Julie Josse**
INRIA
Montpellier, France
`julie.josse@inria.fr`

## Abstract

Stochastic gradient algorithm is a key ingredient of many machine learning methods, particularly appropriate for large-scale learning. However, a major caveat of large data is their incompleteness. We propose an averaged stochastic gradient algorithm handling missing values in linear models. This approach has the merit to be free from the need of any data distribution modeling and to account for heterogeneous missing proportion. In both streaming and finite-sample settings, we prove that this algorithm achieves convergence rate of $\mathcal{O}(\frac{1}{n})$ at the iteration $n$, the same as without missing values. We show the convergence behavior and the relevance of the algorithm not only on synthetic data but also on real data sets, including those collected from medical register.

## 1 Introduction

Stochastic gradient algorithms (SGD) [20] play a central role in machine learning problems, due to their cheap computational cost and memory per iteration. There is a vast literature on its variants, for example using averaging of the iterates [19], some robust versions of SGD [18, 11] or adaptive gradient algorithms like Adagrad [6]; and on theoretical guarantees of those methods [16, 1, 4, 23, 8, 17]. More globally, averaging strategies have been used to stabilize the algorithm behaviour and reduce the impact of the noise, giving better convergence rates without requiring strong convexity.

The problem of missing values is ubiquitous in large scale data analysis. One of the key challenges in the presence of missing data is to deal with the half-discrete nature of the data which can be seen as a mixed of continuous data (observed values) and categorical data (the missing values). In particular for gradient-based methods, the risk minimization with incomplete data becomes intractable and the usual results cannot be directly applied.

**Context.** In this paper, we consider a linear regression model, for $i \geq 1$,

$$y_i = X_{i:}^T \beta^\star + \epsilon_i, \tag{1}$$

parametrized by $\beta^\star \in \mathbb{R}^d$, where $y_i \in \mathbb{R}$, $\epsilon_i \in \mathbb{R}$ is a real-valued centered noise and $X_{i:} \in \mathbb{R}^d$ stands for the real covariates of the $i$-th observation. The $(X_{i:})$'s are assumed to be only partially known, since some covariates may be missing: our objective is to derive stochastic algorithms for estimating the parameters of the linear model, which handle missing data, and come with strong theoretical guarantees on excess risk.

**Related works.** There is a rich literature on handling missing values [13] and yet there are still some challenges even for linear regression models. This is all the more true as we consider such models for large sample size or in high dimension. There are very few regularized versions of regression that can deal with missing values. A classical approach to estimating parameters with

missing values consists in maximizing the observed likelihood, using for instance an Expectation Maximization algorithm [3]. Even if this approach can be implemented to scale for large datasets see for instance [2], one of its main drawbacks is to rely on strong parametric assumptions for the covariates distributions. Another popular strategy to fix the missing values issue consists in predicting the missing values to get a completed data and then in applying the desired method. However matrix completion is a different problem from estimating parameters and can lead to uncontrolled bias and undervalued variance of the estimate [13]. In the regression framework, Jones [10] studied the bias induced by naive imputation.

In the settings of the Dantzig selector [21] and LASSO [14], another solution consists in naively imputing by 0 the incomplete matrix and modifying the algorithm used in the complete case to account for the imputation error. Such a strategy has also been studied by Ma and Needell [15] for SGD in the context of linear regression with missing values and with finite samples: the authors used debiased gradients, in the same spirit as the covariance matrix debiasing considered by Loh and Wainwright [14] in a context of sparse linear regression, or by Koltchinskii et al. [12] for matrix completion. This modified version of the SGD algorithm [15] is conjectured to converge in expectation to the ordinary least squares estimator, achieving the rate of $\mathcal{O}(\frac{\log n}{\mu n})$ at iteration $n$ for the excess empirical risk, assumed to be $\mu$-strongly convex in that work. However, their algorithm requires a step choice relying on the knowledge of the strong-convexity constant $\mu$ which is often intractable for large-scale settings. In a non-linear setting, Yi et al. [25] also propose a heuristic to debias zero-imputation in neural networks but their proposed algorithm comes with no guarantee of convergence.

Besides, the inverse probability weighting method (IPW) consists in keeping *only* complete observations and on reducing the induced bias by reweighting the loss w.r.t. the complete observations with their probabilities of completeness [13, 22]. However, in the IPW literature, weighting is often used to rebalance samples with missing outcome but not in cases where there may be missing values in all covariates, which would imply more complex debiasing expression than simply weighting the data.

**Contributions.**

- We develop a debiased averaged SGD to perform (regularized) linear regression either streaming or with finite samples, when covariates are missing. The approach consists in imputing the covariates with a simple imputation and using debiased gradients accordingly.

- Furthermore, the design is allowed to be contaminated by heterogeneous missing values: each covariate may have a different probability to be missing. This encompasses the classical homogeneous Missing Completely At Random (MCAR) case, where the missingness is independent of any covariate value.

- This algorithm comes with theoretical guarantees: we establish convergence in terms of generalization risk at the rate $1/n$ at iteration $n$. This rate is remarkable as it is (i) optimal w.r.t. $n$, (ii) free from any bad condition number (no strong convexity constant is required), and (iii) similar to the rate of averaged SGD without any missing value. The same convergence rate is also obtained when the probabilities that variables are missing are not known but estimated.

- In terms of performance with respect to the missing entries proportion in large dimension, our strategy results in an error provably several orders of magnitude smaller than the best possible algorithm that would only rely on complete observations.

- We show the relevance of the proposed approach and its convergence behavior on numerical applications and its efficiency on real data; including the TraumaBase®️ dataset to assist doctors in making real-time decisions in the management of severely traumatized patients. The code to reproduce all the simulations and numerical experiments is available on `https://github.com/AudeSportisse/SGD-NA`.

## 2 Problem setting

In this paper, we consider either the streaming setting, i.e. when the data comes in as it goes along, or the finite-sample setting, i.e. when the data size is fixed and form a finite design matrix $X = (X_{1:}| \ldots |X_{n:})^T \in \mathbb{R}^{n \times d}$ ($n > d$). We define $\mathcal{D}_n := \sigma\left((X_{i:}, y_i), i = 1, \ldots, n\right)$ the $\sigma$-field generated by $n$ observations. We also denote $\preccurlyeq$ the partial order between self-adjoint operators, such that $A \preccurlyeq B$ if $B - A$ is positive semi-definite.

Given observations as in (1) and defining $f_i(\beta) := \left(\langle X_{i:}, \beta \rangle - y_i\right)^2 / 2$, the (unknown) linear model parameter satisfies:

$$\beta^\star = \underset{\beta \in \mathbb{R}^d}{\arg\min} \left\{ R(\beta) := \mathbb{E}_{(X_{i:}, y_i)} [f_i(\beta)] \right\}, \tag{2}$$

where $\mathbb{E}_{(X_{i:}, y_i)}$ denotes the expectation over the distribution of $(X_{i:}, y_i)$ (which is independent of $i$ as the observations are assumed to be i.i.d.).

In this work, the covariates are assumed to contain missing values, so one in fact observes $X_{i:}^{\mathrm{NA}} \in (\mathbb{R} \cup \{\mathtt{NA}\})^d$ instead of $X_{i:}$, as $X_{i:}^{\mathrm{NA}} := X_{i:} \odot D_{i:} + \mathtt{NA}(\mathbf{1}_d - D_{i:})$, where $\odot$ denotes the element-wise product, $\mathbf{1}_d \in \mathbb{R}^d$ is the vector filled with ones and $D_{i:} \in \{0, 1\}^d$ is a binary vector mask coding for the presence of missing entries in $X_{i:}$, i.e. $D_{ij} = 0$ if the $(i, j)$-entry is missing in $X_{i:}$, and $D_{ij} = 1$ otherwise. We adopt the convention $\mathtt{NA} \times 0 = 0$ and $\mathtt{NA} \times 1 = \mathtt{NA}$. We consider a *heterogeneous* MCAR setting, i.e. $D$ is modeled with a Bernoulli distribution

$$D = (\delta_{ij})_{1 \le i \le n, 1 \le j \le d} \quad \text{with} \quad \delta_{ij} \sim \mathcal{B}(p_j), \tag{3}$$

with $1 - p_j$ the probability that the $j$-th covariate is missing.

The considered approach consists in imputing the incomplete covariates by zero in $X_{i:}^{\mathrm{NA}}$, as $\tilde{X}_{i:} = X_{i:}^{\mathrm{NA}} \odot D_{i:} = X_{i:} \odot D_{i:}$, and in accounting for the imputation error in the subsequent algorithm.

## 3  Averaged SGD with missing values

The proposed method is detailed in Algorithm 1. The impact of the naive imputation by 0 directly translates into a bias in the gradient. Consequently, at each iteration we use a debiased estimate $\tilde{g}_k$. In order to stabilize the stochastic algorithm, we consider the Polyak-Ruppert [19] averaged iterates $\bar{\beta}_k = \frac{1}{k+1} \sum_{i=0}^{k} \beta_i$.

**Lemma 1.** *Let $(\mathcal{F}_k)_{k \ge 0}$ be the following $\sigma$-algebra, $\mathcal{F}_k = \sigma(X_{1:}, y_1, D_{1:}, \ldots, X_{k:}, y_k, D_{k:})$. The modified gradient $\tilde{g}_k(\beta_{k-1})$ in Equation (4) is $\mathcal{F}_k$-measurable and a.s.,*

$$\mathbb{E}\left[\tilde{g}_k(\beta_{k-1}) \,|\, \mathcal{F}_{k-1}\right] = \nabla R(\beta_{k-1}).$$

**Algorithm 1** Averaged SGD for Heterogeneous Missing Data

---

**Input:** data $\tilde{X}, y, \alpha$ (step size)
Initialize $\beta_0 = 0_d$.
Set $P = \mathrm{diag}\left((p_j)_{j \in \{1, \ldots, d\}}\right) \in \mathbb{R}^{d \times d}$.
**for** $k = 1$ **to** $n$ **do**

$$\tilde{g}_k(\beta_k) = P^{-1} \tilde{X}_{k:} \left( \tilde{X}_{k:}^T P^{-1} \beta_k - y_k \right)$$
$$- (\mathrm{I} - P) P^{-2} \mathrm{diag}\left( \tilde{X}_{k:} \tilde{X}_{k:}^T \right) \beta_k \quad (4)$$

$$\beta_k = \beta_{k-1} - \alpha \tilde{g}_k(\beta_{k-1})$$
$$\bar{\beta}_k = \frac{1}{k+1} \sum_{i=0}^{k} \beta_i = \frac{k}{k+1} \bar{\beta}_{k-1} + \frac{1}{k+1} \beta_k$$
**end for**

---

Lemma 1 is proved in Appendix S2.1. Note that in the case of homogeneous MCAR data, i.e. $p_1 = \ldots = p_d = p \in (0, 1)$, the chosen direction at iteration $k$ in Equation (4) boils down to $\frac{1}{p} \tilde{X}_{k:} \left( \frac{1}{p} \tilde{X}_{k:}^T \beta_k - y_k \right) - \frac{1-p}{p^2} \mathrm{diag}\left( \tilde{X}_{k:} \tilde{X}_{k:}^T \right) \beta_k$, where $\mathrm{diag}(A) \in \mathbb{R}^{d \times d}$ denotes the diagonal matrix containing either the diagonal of $A$ if $A \in \mathbb{R}^{d \times d}$ or the vector $A$ if $A \in \mathbb{R}^d$. This meets the classical debiasing terms of covariance matrices [14, 15, 12]. Note also that in the presence of complete observations, meaning that $p = 1$, Algorithm 1 matches the standard least squares stochastic algorithm.

**Remark 1** (Ridge regularization) Instead of minimizing the theoretical risk as in (2), we can consider a Ridge regularized formulation: $\min_{\beta \in \mathbb{R}^d} R(\beta) + \lambda \|\beta\|^2$, with $\lambda > 0$. Algorithm 1 is trivially extended to this framework: the debiasing term is not modified since the penalization term does not involve the incomplete data $\tilde{X}_{i:}$. This is useful in practice as no implementation is available for incomplete ridge regression.

**Remark 2** (Towards a more general MCAR setting) Note that we consider a specific MCAR setting in Equation (3) in which the missing-data patterns were independent ($D_{.j} \perp\!\!\!\perp D_{.j'}, j \ne j'$). However, an extended MCAR setting could allow coordinates of the missing mask to be dependently missing. In such a case, we propose a new way of constructing debiased versions of gradients, as $\tilde{g}_k(\beta) := (W \odot (\tilde{X}_{k:} \tilde{X}_{k:}^T))\beta - y_k P^{-1} \tilde{X}_{k:}$ with $W \in \mathbb{R}^{d \times d}$, and $W_{ij} := 1/\mathbb{E}[\delta_{ki} \delta_{kj}]$ for $1 \le i, j \le d$. Regarding practical implementation, the matrix $W$ can be estimated, in particular using low-rank strategies on the missing pattern matrix.

# 4 Theoretical results

In this section, we prove convergence guarantees for Algorithm 1 in terms of theoretical excess risk, in both the streaming and the finite-sample settings. For the rest of this section, assume the following.

- The observations $(X_{k:}, y_k) \in \mathbb{R}^d \times \mathbb{R}$ are independent and identically distributed.
- $\mathbb{E}[\|X_{k:}\|^2]$ and $\mathbb{E}[\|y_k\|^2]$ are finite.
- Let $H$ be an invertible matrix, defined by $H := \mathbb{E}_{(X_{k:}, y_k)}[X_{k:} X_{k:}^T]$.

The main technical challenge to overcome is proving that the noise in play due to missing values is *strutured* and still allows to derive convergence results for a debiased version of averaged SGD. This work builds upon the analysis made by Bach and Moulines [1] for standard SGD strategies.

## 4.1 Technical results

Bach and Moulines [1] proved that for least-squares regression, averaged SGD converges at rate $n^{-1}$ after $n$ iterations. In order to derive similar results, we prove in addition to Lemma 1, Lemmas 2 and 3:

- Lemma 2 shows that the noise induced by the imputation by zeros and the subsequent transformation results is a *structured noise*. This is the most challenging part technically: having a structured noise is fundamental to obtain convergence rates scaling as $n^{-1}$ – in the unstructured case the convergence speed is only $n^{-1/2}$ [4].
- Lemma 3 shows that the adjusted random gradients $\tilde{g}_k(\beta)$ are almost surely *co-coercive* [26] i.e., for any $k$, there exists a random "primitive" function $\tilde{f}_k$ which is a.s. convex and smooth, and such that $\tilde{g}_k = \nabla \tilde{f}_k$. Proving that $\tilde{f}_k$ is a.s. convex is an important step which was missing in the analysis of Ma and Needell [15].

**Lemma 2.** *The additive noise process $(\tilde{g}_k(\beta^\star))_k$ with $\beta^\star$ defined in (2) is $\mathcal{F}_k-$measurable and,*

*1. $\forall k \geq 0$, $\mathbb{E}[\tilde{g}_k(\beta^\star) \mid \mathcal{F}_{k-1}] = 0$ a.s..*

*2. $\forall k \geq 0$, $\mathbb{E}[\|\tilde{g}_k(\beta^\star)\|^2 \mid \mathcal{F}_{k-1}]$ is a.s. finite.*

*3. $\forall k \geq 0$, $\mathbb{E}[\tilde{g}_k(\beta^\star)\tilde{g}_k(\beta^\star)^T] \preccurlyeq C(\beta^\star) = c(\beta^\star)H$, with*

$$c(\beta^\star) = \frac{\mathrm{Var}(\epsilon_k)}{p_m^2} + \left(\frac{(2 + 5p_m)(1 - p_m)}{p_m^3}\right)\gamma^2\|\beta^\star\|^2. \tag{5}$$

*Sketch of proof (Lemma 2).* Property 1 easily followed from Lemma 1 and the definition of $\beta^\star$. Property 2 can be obtained with similar computations as in [15, Lemma 4]. Property 3 cannot be directly derived from Property 2, since $\tilde{g}_k(\beta^\star)\tilde{g}_k(\beta^\star)^T \preccurlyeq \|\tilde{g}_k(\beta^\star)\|^2 I$ leads to an insufficient upper bound. Proof relies on decomposing the external product $\tilde{g}_k(\beta^\star)\tilde{g}_k(\beta^\star)^T$ in several terms and obtaining the control of each, involving technical computations. $\square$

**Lemma 3.** *For all $k \geq 0$, given the binary mask $D$, the adjusted gradient $\tilde{g}_k(\beta)$ is a.s. $L_{k,D}$-Lipschitz continuous, i.e. for all $u, v \in \mathbb{R}^d$, $\|\tilde{g}_k(u) - \tilde{g}_k(v)\| \leq L_{k,D}\|u - v\|$ a.s.. Set*

$$L := \sup_{k,D} L_{k,D} \leq \frac{1}{p_m^2}\max_k \|X_{k:}\|^2 \ a.s.. \tag{6}$$

*In addition, for all $k \geq 0$, $\tilde{g}_k(\beta)$ is almost surely co-coercive.*

Lemmas 2 and 3 are respectively proved in Appendices S2.2 and S2.3, and can be combined with Theorem 1 in [1] in order to prove the following theoretical guarantees for Algorithm 1.

## 4.2 Convergence results

The following theorem quantifies the convergence rate of Algorithm 1 in terms of excess risk.

**Theorem 1** (Streaming setting). *Assume that for any $i$, $\|X_{i:}\| \leq \gamma$ almost surely for some $\gamma > 0$. For any constant step-size $\alpha \leq \frac{1}{2L}$, Algorithm 1 ensures that, for any $k \geq 0$:*

$$\mathbb{E}\left[R\left(\bar{\beta}_k\right) - R(\beta^\star)\right] \leq \frac{1}{2k}\left(\frac{\sqrt{c(\beta^\star)d}}{1-\sqrt{\alpha L}} + \frac{\|\beta_0 - \beta^\star\|}{\sqrt{\alpha}}\right)^2,$$

*with $L$ given in Equation (6), $p_m = \min_{j=1,\ldots d} p_j$ and $c(\beta^\star)$ given in Equation (5).*

Note that in Theorem 1, the expectation is taken over the randomness of the observations $(X_{i:}, y_i, D_{i:})_{1 \leq i \leq k}$. The bounded features assumption in Theorem 1 is mostly convenient for the readability, but it can be relaxed at the price of milder but more technical assumptions and proofs (typically bounds on quadratic mean instead of a.s. bounds, see e.g. Section 6.1. in [5]).

**Remark 3** (Finite-sample setting) Similar results as Theorem 1 can be derived in the case of finite-sample setting. For the sake of clarity, they are made explicit hereafter: for any constant step-size $\alpha \leq \frac{1}{2L}$, Algorithm 1 ensures that for any $k \leq n$:
$\mathbb{E}\left[R(\bar{\beta}_k) - R(\beta^\star)\right]|\mathcal{D}_n] \leq \frac{1}{2k}\left(\frac{\sqrt{c(\beta^\star)d}}{1-\sqrt{\alpha L}} + \frac{\|\beta_0-\beta^\star\|}{\sqrt{\alpha}}\right)^2$ with $L$ given in Equation (6) and $c(\beta^\star) = \frac{\mathrm{Var}(\epsilon_k)}{p_m^2} + \left(\frac{(2+5p_m)(1-p_m)}{p_m^3}\right)\max_{1\leq i \leq n}\|X_{i:}\|^2\|\beta^\star\|^2$.

**Remark 4** (Estimating missing probabilities $(\hat{p}_j)_j$) Algorithm 1 and the associated convergence rate established in Theorem 3 require the knowledge of the missing probabilities $(p_j)_j$. In practice, one could construct an estimator $\hat{\bar{\beta}}_k$ using our algorithm with estimated probabilities $(\hat{p}_j)_j$. In such a case, we can show that we preserve the convergence rate at $1/k$. More precisely, in the finite-sample setting, we can use the first half of the data to evaluate the $(\hat{p}_j)_j$'s and the second half of the data to build $\hat{\bar{\beta}}_k$. Under the additional assumptions of bounded iterates and strong convexity of the risk, the resulting supplementary risk w.r.t. the iterate $\bar{\beta}_k$ built with the true $(p_j)_j$ is $\mathbb{E}[R(\hat{\bar{\beta}}_k) - R(\bar{\beta}_k)] = \mathcal{O}(1/kp_{\min}^6)$. This is formalized in Theorem 2 of Appendix S3, followed by its proof.

**Convergence rates for the iterates.** Note that if a Ridge regularization is considered, the regularized function to minimize $R(\beta)+\lambda\|\beta\|^2$ is $2\lambda$-strongly convex. Theorem 1 and Remark 3 then directly provide the following bound on the iterates: $\mathbb{E}\left[\left\|\overline{\beta}_k - \beta^\star\right\|^2\right] \leq \frac{1}{2\lambda k}\left(\frac{\sqrt{c(\beta^\star)d}}{1-\sqrt{\alpha L}} + \frac{\|\beta_0-\beta^\star\|}{\sqrt{\alpha}}\right)^2$.

**Additional comments.** We highlight the following points:

- In Theorem 1, the expected excess risk is upper bounded by (a) *a variance* term, that grows with the noise variance and is increased by the missing values, and (b) *a bias* term, that accounts for the importance of the initial distance between the starting point $\beta_0$ and the optimal one $\beta^\star$.

- The optimal convergence rate is achieved for a *constant* learning rate $\alpha$. One could for example choose $\alpha = \frac{1}{2L}$, that does *not decrease* with the number of iterations. In such a situation, both the *bias* and *variance* terms scale as $k^{-1}$. Remark that convergence of the averaged SGD with constant step-size only happens for least squares regression, because the un-averaged iterates converge to a limit distribution whose mean is exactly $\beta^*$ [1, 5].

- The expected risk scales as $n^{-1}$ after $n$ iterations, without strong convexity constant involved.

- For the generalization risk $R$, this rate of $n^{-1}$ is known to be statistically optimal for least-squares regression: under reasonable assumptions, no algorithm, even more complex than averaged SGD or without missing observations, can have a better dependence in $n$ [24].

- In the complete case, i.e. when $p_1 = \ldots = p_d = 1$, Theorem 1 and Remark 3 meet the results from Bach and Moulines [1, Theorem 1]. Indeed, in such a case, $c(\beta^\star) = \mathrm{Var}(\epsilon_k)$.

- The noise variance coefficient $c(\beta^\star)$ includes (i) a first term as a classical noise one, proportional to the model variance, and increased by the missing values occurrence to $\frac{\mathrm{Var}(\epsilon_k)}{p_m^2}$; (ii) the second term is upper-bounded by $\frac{7(1-p_m)}{p_m^3}\cdot\gamma^2\|\beta^\star\|^2$ corresponds to the multiplicative noise induced by the imputation by 0 and gradient debiasing. It naturally increases as the radius $\gamma^2$ of the observations increases (so does the imputation error), and vanishes if there are no missing values ($p_m = 1$).

**Remark 5** (Only one epoch) It is important to notice that in a finite-sample setting, as covered by Remark 3, given a maximum number of $n$ observations, our convergence rates are only valid for $k \leq n$: the theoretical bound holds only for one pass on the input/output pairs. Indeed, afterwards, we cannot build unbiased gradients of the risk.

## 4.3 What about empirical risk minimization (ERM)?

**Theoretical locks.** Note that the translation of the results in Remark 3 in terms of empirical risk convergence is still an open issue. The heart of the problem is that it seems really difficult to obtain a sequence of unbiased gradients of the empirical risk.

- Indeed, to obtain unbiased gradients, the data should be processed *only once* in Algorithm 1: if we consider the gradient of the loss with respect to an observation $k$, we obviously need the binary mask $D_k$ and the current point $\beta_{k-1}$ to be independent for the correction relative to the missing entries to make sense. As a consequence, no sample can be used twice - in fact, running multiple passes over a finite sample could result in over-fitting the missing entries.

- Therefore, with a finite sample at hand, the sample used at each iteration should be chosen *without replacement* as the algorithm runs. But even in the complete data case, sampling without replacement induces a bias on the chosen direction [7, 9]. Consequently, Lemma 1 does not hold for the empirical risk instead of the theoretical one. This issue is not addressed in [15], unfortunately making the proof of their result invalid/wrong.

**Comparison to Ma and Needell [15].** Leaving aside the last observation, we can still comment on the bounds in [15] for the empirical risk without averaging. As they do not use averaging but only the last iterate, their convergence rate (see Lemma 1 in their paper) is only studied for $\mu-$strongly convex problems and is expected to be larger (i) by a factor $\mu^{-1}$, due to the choice of their decaying learning rate, and (ii) by a $\log n$ factor due to using the last iterate and not the averaged one [23]. Moreover, the strategy of the present paper does not require to access the strong convexity constant, which is generally out of reach, if no explicit regularization is used. More marginally, we provide the proof of the co-coercivity of the adjusted gradients (Lemma 3), which is required to derive the convergence results, and which was also missing in Ma and Needell [15]. A more detailed discussion on the differences between the two papers is given in Appendix S1.

**ERM hindered by NA.** It is also interesting to point out that with missing features, *neither* the generalization risk $R$, *nor* the empirical risk $R_n$ are observed (i.e., only approximations of their values or gradients can be computed). As a consequence, one cannot expect to minimize those functions with unlimited accuracy. This stands in contrast to the *complete observations setting*, in which the empirical risk $R_n$ is known exactly. As a consequence, with missing data, empirical risk loses its main asset - being an observable function that one can minimize with high precision. Overall it is both more natural and easier to focus on the generalization risk.

## 4.4 On the impact of missing values

**Marginal values of incomplete data.** An important question in practice is to understand how much information has been lost because of the incompleteness of the observations. In other words, it is better to access 200 input/output pairs with a probability 50% of observing each feature on the inputs, or to observe 100 input/output pairs with complete observations?

Without missing observations, the variance bound in the expected excess risk is given by Theorem 1 with $p_m = 1$: it scales as $\mathcal{O}\left(\frac{\mathrm{Var}(\epsilon_k)d}{k}\right)$, while with missing observations it increases to $\mathcal{O}\left(\frac{\mathrm{Var}(\epsilon_k)d}{kp_m^2} + \frac{C(X,\beta^\star)}{kp_m^3}\right)$. As a consequence, the variance upper bound is larger by a factor $p_m^{-1}$ for the estimator derived from $k$ incomplete observations than for $k \times p_m$ complete observations. This suggests that there is a higher gain to collecting fewer complete observations (e.g., 100) than more incomplete ones (e.g., 200 with $p = 0.5$). However, one should keep in mind that this observation is made by comparing upper bounds thus does not necessarily reflect what would happen in practice.

**Keeping only complete observations?** Another approach to solve the missing data problem is to discard all observations that have at least one missing feature. The probability that one input

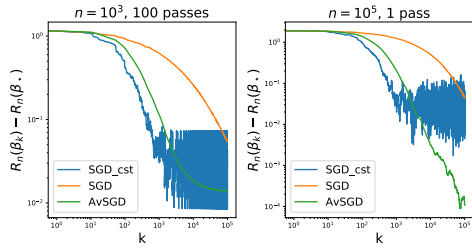
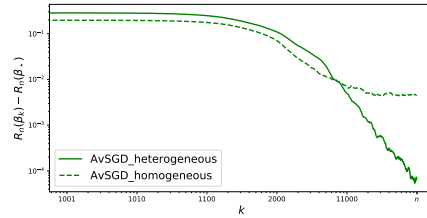

Figure 1: Empirical excess risk $(R_n(\beta_k) - R_n(\beta^\star))$. Left: $n = 10^3$ and 100 passes. Right: $n = 10^5$ and 1 pass. $d = 10$, 30% MCAR data. $L$ is assumed to be known in both graphics.

Figure 2: Empirical excess risk $R_n(\beta_k) - R_n(\beta^\star)$ for synthetic data where $n = 10^5$, $d = 10$ and with heterogeneous missing values either taking into account the heterogeneity (plain line) in the algorithm or not (dashed line).

is complete, under our missing data model is $\prod_{j=1}^{d} p_j$. In the homogeneous case, the number of complete observations $k_{co}$ out of a $k-$sample thus follows a binomial law $k_{co} \sim \mathcal{B}(k, p^d)$. With only those few observations, the statistical lower bound is $\mathrm{Var}(\epsilon_k)d/k_{co}$. In expectation, by Jensen inequality, we get that the lower bound on the risk is larger than $\mathrm{Var}(\epsilon_k)d/kp^d$.

Our strategy thus leads to an *upper-bound* which is typically $p^{d-3}$ times smaller than the *lower bound* on the error of any algorithm relying only on complete observations. For a large dimension or a high percentage of missing values, our strategy is thus provably several orders of magnitude smaller than the best possible algorithm that would only rely on complete observations - e.g., if $p = 0.9$ and $d = 40$, the error of our method is at least 50 times smaller.

Also note that in Theorem 1 and Lemma 1 in Ma and Needell [15], the convergence rate with missing observations suffers from a similar multiplicative factor $\mathcal{O}(p^{-2} + \kappa p^{-3})$.

## 5 Experiments

### 5.1 Synthetic data

Consider the following simulation setting: the covariates are normally distributed, $X_{i:} \overset{i.i.d.}{\sim} \mathcal{N}(0, \Sigma)$, where $\Sigma$ is constructed using uniform random orthogonal eigenvectors and decreasing eigenvalues $1/k$, $k = 1, \ldots, d$. For a fixed parameter vector $\beta$, the outputs $y_i$ are generated according to the linear model (1), with $\epsilon_i \sim \mathcal{N}(0, 1)$. Setting $d = 10$, we introduce 30% of missing values either with a uniform probability $p$ of missingness for any feature, or with probability $p_j$ for covariate $j$, with $j = 1, \ldots, d$. Firstly, the three following algorithms are implemented:

(1) **AvSGD** described in Algorithm 1 with a constant step size $\alpha = \frac{1}{2L}$, and $L$ given in (6).

(2) **SGD** from [15] with iterates $\beta_{k+1} = \beta_k - \alpha_k \tilde{g}_{i_k}(\beta_k)$, and decreasing step size $\alpha_k = \frac{1}{\sqrt{k+1}}$.

(3) **SGD_cst** from [15] with a constant step size $\alpha = \frac{1}{2L}$, where $L$ is given by (6).

**Debiased averaged vs. standard SGD.** Figure 1 compares the convergence of Algorithms (1), (2) and (3), with either multiple passes or one pass, in terms of excess empirical risk $R_n(\beta) - R_n(\beta^\star)$, with $R_n(\beta) := \frac{1}{n} \sum_{i=1}^{n} f_i(\beta)$. As expected (see Remark 5 and Subsection 4.3), multiple passes can lead to saturation: after one pass on the observations, AvSGD does not improve anymore (Figure 1, left), while it keeps decreasing in the streaming setting (Figure 1, right). Looking at Figure 1 (right), one may notice that without averaging and with decaying step-size, Algorithm (2) achieves the convergence rate $\mathcal{O}\left(\sqrt{\frac{1}{n}}\right)$, whereas with constant step-size, Algorithm (3) saturates at an excess risk proportional to $\alpha$ after $n = 10^3$ iterations. As theoretically expected, both methods are improved with averaging. Indeed, Algorithm 1 converges pointwise with a rate of $\mathcal{O}(\frac{1}{n})$.

**About the algorithm hyperparameter.** Note that the Lipschitz constant $L$ given in (6) can be either computed from the complete covariates, or estimated from the incomplete data, see discussion and numerical experiments in Appendix S4.

**Heterogeneous vs. homogeneous missingness.** In Figure 2, the missing values are introduced with different missingness probabilities, i.e. with distinct $(p_j)_{1 \le j \le d}$ per feature, as described in Equation (3). When taking into account this heterogeneousness, Algorithm 1 achieves the same convergence rates as in Figure 1. However, ignoring the heterogeneous probabilities in the gradient debiasing leads to stagnation far from the optimum in terms of empirical excess risk.

**Increasing missing data proportions.** Figure 3 shows the results of Algorithm 1 with different percentage of missing values (25%, 50% and 75%). The more missing data there are, the more the convergence rate deteriorates. This was expected, as the established theoretical upper bound for the convergence in Theorem 1 increases as the probability of being observed gets smaller.

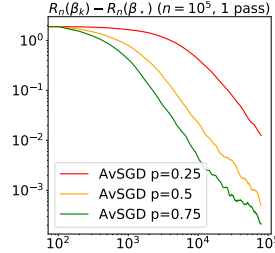

**Polynomial features.** Algorithm 1 can be adapted to handle missing polynomial features, see Appendix S5 for a detailed discussion and numerical experiments on synthetic data.

Figure 3: Empirical excess risk $R_n(\beta_k) - R_n(\beta^\star)$ for synthetic data where $n = 10^5$, $d = 10$ with 25% (green), 50% (orange) and 75% (red) missing values.

## 5.2 Real dataset 1: Traumabase dataset

We illustrate our approach on a public health application with the APHP TraumaBase® Group (Assistance Publique - Hopitaux de Paris) on the management of traumatized patients. Our aim is to model the level of platelet upon arrival at the hospital from the clinical data of 15785 patients. The platelet is a cellular agent responsible for clot formation and it is essential to control its levels to prevent blood loss and to decide on the most suitable treatment. A better understanding of the impact of the different features is key to trauma management. Explanatory variables for the level of platelet consist in seven quantitative (missing) variables, which have been selected by doctors. In Figure 4, one can see the percentage of missing values in each variable, varying from 0 to 16%, see Appendix S6 for more information on the data.

**Model estimation.** The model parameter estimation is performed either using the AvSGD Algorithm 1 or an Expectation Maximization (EM) algorithm [3]. Both methods are compared with the ordinary least squares linear regression in the complete case, i.e. keeping the fully-observed rows only (i.e. 9448 rows). The signs of the coefficients for Algorithm 1 are shown in Figure 4.

According to the doctors, a negative effect of shock index (*SI*), vascular filling (*VE*), blood transfusion (*RBC*) and lactate (*Lactacte*) was expected, as they all result in low platelet levels and therefore a higher risk of severe bleeding. However, the effects of delta Hemocue (*Delta.Hemocue*) and the heart rate (*HR*) on platelets are not entirely in agreement with their opinion. Note that using the linear regression in the complete case and the EM algorithm lead to the same sign for the variables effects as presented in Figure 4.

## 5.3 Real dataset 2: Superconductivity dataset

We now consider the Superconductivity dataset (available here), which contains 81 quantitative features from 21263 superconductors. The goal here is to predict the critical temperature of each superconductor. Since the dataset is initially complete, we introduce 30% of missing values with probabilities $(p_j)_{1 \le j \le 81}$ for the covariate $j$, with $p_j$ varying between 0.7 and 1. The results are shown in Figure 5 where a Ridge regularization has been added or not. The regularization parameter $\lambda$ (see Remark 1) is chosen by cross validation.

**Prediction performance.** The dataset is divided into training and test sets (random selection of $70 - 30\%$). The test set does not contain missing values. In order to predict the critical temperature of each superconductor, we compute $\hat{y}_{n+1} = X_{n+1}^T \hat{\beta}$ with $\hat{\beta} = \beta_n^{\text{AvSGD}}$ or $\beta_n^{\text{EM}}$. We also impute

| Variable | Effect | NA % |
|----------|--------|------|
| Lactate | − | 16% |
| $\Delta$.Hemo | + | 16% |
| VE | − | 9% |
| RBC | − | 8% |
| SI | − | 2% |
| HR | + | 1% |
| Age | − | 0% |

Figure 4: Percentage of missing features, and effect of the variables on the platelet for the TraumaBase data when the AvSGD algorithm is used. "+" indicates positive effect while "−" negative.

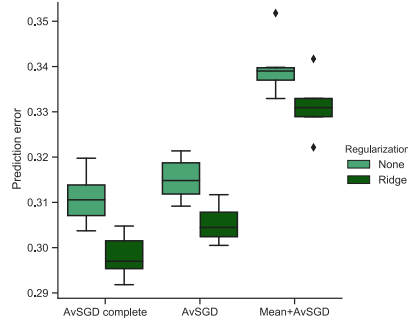

Figure 5: Prediction error boxplots (over 10 replications) for the Superconductivity data. AvSGD complete corresponds to applying the AvSGD on the complete data, AvSGD and Mean+AvSGD use the predictions obtained with the estimated parameters $\hat{\beta}_n^{\text{AvSGD}}$ and $\bar{\beta}_n^{\text{AvSGD}}$ respectively.

the missing data naively by the mean in the training set, and apply the averaged stochastic gradient without missing data on this imputed dataset, giving a coefficient model $\bar{\beta}_n^{\text{AvSGD}}$. It corresponds to the case where the bias of the imputation has not been corrected. The prediction quality on the test set is compared according to the relative $\ell_2$ prediction error, $\|\hat{y} - y\|^2/\|y\|^2$. The data is scaled, so that the naive prediction by the mean of the outcome variable leads to a prediction error equal to 1. In Figure 5, we observe that the SGD strategies give quite good prediction performances. The EM algorithm is not represented since it is completely out of range (the mean of its prediction error is 0.7), which indicates that it struggles with a large number of covariates. Note also that the EM algorithm requires a distributional assumption on the covariates, which is not the case of our method. As for the AvSGD Algorithm, it performs well in this setting. Indeed, with or without regularization, the prediction error with missing values is very close to the one obtained from the complete dataset.

Algorithm 1 is shown to handle missing polynomial features well even in higher dimensions, see Appendix S5 for a detailed discussion and large-scale experiments on the superconductivity dataset.

**Comparison to other methods.** For completeness, we ran the proposed algorithm on the superconductivity dataset and compare it to two-step heuristics in which first, the covariates are imputed (by the mean or by the ICE[1] iterative imputer that estimates each feature from all the others) and then linear regression (LR) is performed on the completed data. The coefficient of determination $R^2$ is plotted on Figure 6 (thus higher is better) for the Superconductivity dataset with 60% of missing values. Our method greatly outperforms all other methods, and follows closely the linear regression performed on the initial complete data. One should note that the two-step heuristics considered here come with no theoretical guarantee at all.

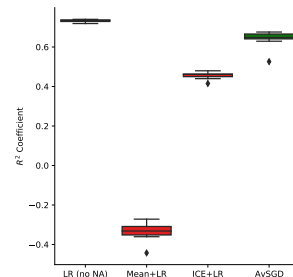

Figure 6: $R^2$ coefficients for the Superconductivity data with 60% MCAR values.

# 6 Discussion

In this work, we thoroughly study the impact of missing values for Stochastic Gradient Descent algorithm for Least Squares Regression. We leverage both the power of averaging and a simple and powerful debiasing approach to derive tight and rigorous convergence guarantees for the generalization risk of the algorithm. The theoretical study directly translates into practical recommendations for the users and a byproduct is the availability of a python implementation of regularized regression with missing values for large scale data, which was not available. Even though we have knocked down some barriers, there are still exciting perspectives to be explored as the robustness of the approach to rarely-occurring covariates, or dealing with more general loss functions as well - for which it is challenging to build a debiased gradient estimator from observations with missing values, or also considering more complex missing-data patterns such as missing-not-at-random mechanisms.

## Funding disclosure

The work was partly supported by the chaire SCAI (ANR-19-CHIA-0002-0).

## Broader impact

Our goal is to provide a solid and rigorous theoretical understanding, in simple enough situations, of what we can achieve with missing data, together with optimal algorithms. Being able to reduce the burden of missing entries in datasets can avoid the unnecessary effort of collecting new complete datasets, and facilitate learning in situations in which data has been collected from many different sources, without the possibility of a centralized coordination. This is typically the case in medical domains, in which different hospitals, or countries, typically gather different observations on the patients. This is why we evaluated its efficiency on real dataset of the medical register TraumaBase.

The use of stochastic algorithms, which are widespread and crucial for large-scale learning allows us to focus on the generalization error, reducing the risk of overfitting.

As missing data are ubiquitous in machine learning, and this work is not directly targeted at any type of applications, its impact is inherently dependent on the domain in which it is used.

## Footnotes

[1] `sklearn.impute.IterativeImputer`

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
