[Supplementary Material]

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

## S1    Discussion on the paper of Ma and Needell [15]

In this section, we make the theoretical issues unlocked in Ma and Needell [15] explicit. For clarity, we directly refer to the lemmas and theorems as numbered in the published version (http://www. global-sci.org/uploads/online_news/NMTMA/201809051633-2442.pdf), the numbering being slightly different than the arXiv version. For readability, we translate their method and

results with the notation used in the present paper. In their paper, they consider the finite-sample setting, with at hand $(D_{i:}, \tilde{X}_{i:})_{1 \leq i \leq n}$, in view of minimizing the empirical risk.

As a preamble, let us remind that the contributions of the present paper go far beyond correcting the approach in [15]: we propose a different algorithm using averaging, that converges faster and in a non-strongly convex regime, with a different proof technique, requiring a more technical proof on the second order moment of the noise, and we allow for heterogeneity in the missing data mechanism.

### S1.1   Hurdles to get unbiased gradients of the empirical risk

The stochastic gradients in [15] are not unbiased gradients of the empirical risk (which makes their main result wrong). Indeed, their algorithm uses the debiased direction (4) by sampling uniformly with replacement the $(\tilde{X}_{k:})_k$'s.

For clarity, we highlight both why the result is not technically correct in their paper, and why it is not intuitively possible to achieve the result they give.

**Technically.** The proof of the main Theorem 2.2 (Theorem 2.1 being a direct corollary), corresponds to the classical proof in which one upper bounds the expectation of the mean-squared distance from the iterate at iteration $k + 1$ to the optimal point **conditionally to the iterate at iteration** $k$, or more precisely, conditionally to a $\sigma$-algebra making this iterate measurable. This is typically written

$$\mathbb{E}\left[||\beta_{k+1} - \beta_*^n||^2 | \mathcal{F}_k\right],$$

where $\beta_*^n$ is the minimizer of the empirical risk $R_n$ and $\beta_k$ is $\mathcal{F}_k$-measurable.

The crux of the proof is then to use **unbiased gradients conditionally to** $\mathcal{F}_k$: the property needed is that

$$\mathbb{E}\left[g_{i_{k+1}}(\beta_k) | \mathcal{F}_k\right] = \nabla R_n(\beta_k).$$

In classical ERM (without missing value) it is done by sampling uniformly at iteration $k + 1$ one observation indexed by $i_{k+1} \sim \mathcal{U}[\![1; n]\!]$, **independently from** $\beta_k$.

In regression with missing data, one has to deal with another source of randomness, the randomness of the mask $D$. In Ma and Needell [15], Lemma A.1 states that for a random $i \sim \mathcal{U}[\![1; n]\!]$ and a matrix row $A_i$, for a random mask $D$ associated to this row,

$$\mathbb{E}_D[\mathbb{E}_i g_i(\beta)] = \nabla R_n(\beta).$$

This lemma is valid. Unfortunately, its usage in the proof of Theorem 2.2 (page 18, line (ii)), is not, as one *does not have*:

$$\mathbb{E}[g_{i_{k+1}}(\beta_k) | \mathcal{F}_k] = \nabla R_n(\beta_k),$$

indeed,

- either the sample $i_{k+1}$ is chosen uniformly at random in $[\![1; n]\!]$ and $D_{i_{k+1}}$ **is not** independent from $\beta_k$.
- or the sample $i$ is not chosen uniformly in $[\![1; n]\!]$ (for example without replacement, as we do) and then the gradient is not an unbiased gradient of $R_n$ as the sampling is not uniform anymore.

In other words, the proof would only be valid if **the mask for the missing entries was re-sampled each time the point is used**, which is of course not realistic for a missing data approach (that would mean that the data has in fact been collected without missing entries).

**Intuition on why it is hard.** A way to understand the impossibility of having a bound for multiple pass on ERM in the context of missing data is to underline that the empirical risk, in the presence of missing data, is an **unknown function**: its value cannot be computed exactly (see Subsection 4.3).

As a consequence we can hardly expect that one could minimize it to unlimited accuracy. This is very similar to the situation for the *generalization risk* in a situation *without missing data*: as the function is not observed, it is impossible to minimize it exactly. Given only $n$ observations, no algorithm can achieve 0-generalization error (and statistical lower bounds [24] prove so).

**Conclusion.** This highlights how difficult it is to be rigorous when dealing with multiple sources of randomness. Unfortunately, none of these limits are discussed in the current version of [15].This makes the approach and the main theorem of [15] mathematically invalid. In the present paper, the *generalization risk* is decaying *during the first pass*, and as a consequence, the empirical risk also probably does, but this has not been proved yet.

In the following paragraph, we give details on the missing technical Lemma.

### S1.2 Missing key Lemma in the proof.

Proving that $(\tilde{f}_k)$ is a.s. convex is an important step for convergence, which was missing in the analysis of [15]. More precisely, in Lemma A.4. in [15], a condition is missing on $G(x)$: $G$ needs to be smooth *and convex* for its gradient to satisfy the co-coercivity inequality. Note that this condition was also missing in the paper they refer to Needell et al. [17] (Indeed, at the third line of the proof of Lemma A.1. in Needell et al. [17], one needs $f$ to be convex for $G$ to be convex). Co-coercivity of the gradient is indeed a characterization of the fact that the function is smooth and convex, see for example Zhu and Marcotte [26].

## S2 Proofs of technical lemmas

Recall that we aim at minimizing the theoretical risk in both streaming and finite-sample settings.
$$\beta^\star = \arg\min_{\beta \in \mathbb{R}^d} R(\beta) = \arg\min_{\beta \in \mathbb{R}^d} \mathbb{E}_{(X_{i:},y_i)} \left[ f_i(\beta) \right]. \tag{2}$$

In the sequel, one consider the following modified gradient direction
$$\tilde{g}_k(\beta_k) = P^{-1}\tilde{X}_{k:} \left( \tilde{X}_{k:}^T P^{-1} \beta_k - y_k \right) - (I - P)P^{-2}\text{diag}\left( \tilde{X}_{k:}\tilde{X}_{k:}^T \right) \beta_k. \tag{4}$$

Note that for all $k$, $D_{k:}$ is independent from $(X_{k:}, y_k)$. In what follows, the proofs are derived considering
$$\mathbb{E} = \mathbb{E}_{(X_{k:},y_k),D_{k:}} = \mathbb{E}_{(X_{k:},y_k)}\mathbb{E}_{D_{k:}}$$
where $\mathbb{E}_{(X_{k:},y_k)}$ and $\mathbb{E}_{D_{k:}}$ denotes the expectation with respect to the distribution of $(X_{k:}, y_k)$ and $D_{k:}$ respectively.

### S2.1 Proof of Lemma 1

**Lemma 4.** *Let $(\mathcal{F}_k)_{k \geq 0}$ be the following $\sigma$-algebra,*
$$\mathcal{F}_k = \sigma(X_{1:}, y_1, D_{1:} \ldots, X_{k:}, y_k, D_{k:}).$$
*The modified gradient $\tilde{g}_k(\beta_{k-1})$ in Equation (4) is $\mathcal{F}_k$-measurable and*
$$\mathbb{E}\left[ \tilde{g}_k(\beta_{k-1}) \mid \mathcal{F}_{k-1} \right] = \nabla R(\beta_{k-1}) \quad a.s.$$

*Proof.*
$$\mathbb{E}_{(X_{k:},y_k),D_{k:}} \left[ \tilde{g}_k(\beta_{k-1}) | \mathcal{F}_{k-1} \right]$$
$$\overset{(i)}{=} \mathbb{E}_{(X_{k:},y_k),D_{k:}} \left[ P^{-1}\tilde{X}_{k:}\tilde{X}_{k:}^T P^{-1} \right] \beta_{k-1} - \mathbb{E}_{(X_{k:},y_k),D_{k:}} \left[ P^{-1}\tilde{X}_{k:}y_k \right]$$
$$- \mathbb{E}_{(X_{k:},y_k),D_{k:}} \left[ (I-P)P^{-2}\text{diag}\left( \tilde{X}_{k:}\tilde{X}_{k:}^T \right) \right] \beta_{k-1}$$
$$\overset{(ii)}{=} \mathbb{E}_{(X_{k:},y_k)} \left[ P^{-1}PX_{k:}X_{k:}^T PP^{-1}\beta_{k-1} + P^{-2}(P - P^2)\text{diag}(X_{k:}X_{k:}^T)\beta_{k-1} - P^{-1}PX_{k:}y_k \right]$$
$$- \mathbb{E}_{(X_{k:},y_k)} \left[ (I-P)P^{-2}P\text{diag}\left( X_{k:}X_{k:}^T \right) \beta_{k-1} \right]$$
$$= \nabla R(\beta_{k-1}),$$
In step (i), we use that $\beta_{k-1}$ is $\mathcal{F}_{k-1}$-measurable and $(X_k, y_k, D_k)$ is independent from $\mathcal{F}_{k-1}$. Step (ii) follows from
$$\begin{cases} \mathbb{E}_{D_{k:}} \left[ \tilde{X}_{k:}\tilde{X}_{k:}^T \right] &= PX_{k:}X_{k:}^T P + (P - P^2)\text{diag}(X_{k:}X_{k:}^T), \\ \mathbb{E}_{D_{k:}} \left[ \text{diag}(\tilde{X}_{k:}\tilde{X}_{k:}^T) \right] &= P\text{diag}(X_{k:}X_{k:}^T), \\ \mathbb{E}_{D_{k:}} \left[ \tilde{X}_{k:} \right] &= PX_{k:}. \end{cases}$$

$\square$

## S2.2  Proof of Lemma 2

**Lemma 5.** *The additive noise process $(\tilde{g}_k(\beta^\star))_k$ with $\beta^\star$ defined in Equation (2) is $\mathcal{F}_k-$measurable and has the following properties:*

1. *$\forall k \geq 0$, $\mathbb{E}[\tilde{g}_k(\beta^\star) \mid \mathcal{F}_{k-1}] = 0$ a.s.,*

2. *$\forall k \geq 0$, $\mathbb{E}[\|\tilde{g}_k(\beta^\star)\|^2 \mid \mathcal{F}_{k-1}]$ is a.s. finite,*

3. *$\forall k \geq 0$, $\mathbb{E}[\tilde{g}_k(\beta^\star)\tilde{g}_k(\beta^\star)^T] \preccurlyeq C(\beta^\star) = c(\beta^\star)H$, where $\preccurlyeq$ denotes the order between self-adjoint operators ($A \preccurlyeq B$ if $B - A$ is positive semi-definite).*

*Proof.* 1 The first point is easily verified using Lemma 1 combined with $\nabla R(\beta^\star) = 0$ by (2).

2 Let us first remark that by independence $\mathbb{E}[\|\tilde{g}_k(\beta^\star)\|^2 \mid \mathcal{F}_{k-1}] = \mathbb{E}[\|\tilde{g}_k(\beta^\star)\|^2]$. Then,

$$\mathbb{E}[\|\tilde{g}_k(\beta^\star)\|^2] \leq \frac{1}{p_m^2}\mathbb{E}\left[\|X_{k:}\|^2\left(\tilde{X}_{k:}^T P^{-1}\beta^\star - y_k\right)^2\right] + \frac{(1-p_m)^2}{p_m^2}\mathbb{E}\left[\|P^{-1}\mathrm{diag}\left(\tilde{X}_{k:}\tilde{X}_{k:}^T\right)\beta^\star\|^2\right].$$

We decompose the computation with respect to $\mathbb{E}_{D_{k:}}$ first,

$$
\begin{aligned}
\mathbb{E}_{D_{k:}}\left[\left(\tilde{X}_{k:}^T P^{-1}\beta^\star - y_k\right)^2\right] &= \mathbb{E}_{D_{k:}}\left[(\tilde{X}_{k:}^T P^{-1}\beta^\star)^2\right] - 2y_k\mathbb{E}_{D_{k:}}\left[\tilde{X}_{k:}^T P^{-1}\beta^\star\right] + y_k^2 \\
&= \mathbb{E}_{D_{k:}}\left[\left(\sum_{j=1}^d \tilde{X}_{kj}p_j^{-1}\beta_j^\star\right)^2\right] - 2y_k\mathbb{E}_{D_{k:}}\left[\sum_{j=1}^d \tilde{X}_{kj}p_j^{-1}\beta_j^\star\right] + y_k^2 \\
&= \sum_{j=1}^d \mathbb{E}_{D_{k:}}\left[\tilde{X}_{kj}^2 p_j^{-2}\beta_j^{\star 2}\right] + 2\sum_{l<j}\mathbb{E}_{D_{k:}}\left[\tilde{X}_{kj}\tilde{X}_{kl}p_j^{-1}p_l^{-1}\beta_j^\star\beta_l^\star\right] \\
&\qquad - 2y_k\sum_{j=1}^d X_{kj}\beta_j^\star + y_k^2 \\
&= \sum_{j=1}^d p_j^{-1}X_{kj}^2\beta_j^{\star 2} + 2\sum_{l<j}X_{kj}X_{kl}\beta_j^\star\beta_l^\star - 2y_k\sum_{j=1}^d X_{kj}\beta_j^\star + y_k^2 \\
&= (X_{k:}^T\beta^\star - y_k)^2 + \sum_{j=1}^d (p_j^{-1} - 1)X_{kj}^2\beta_j^{\star 2},
\end{aligned}
$$

which gives

$$\mathbb{E}_{D_{k:}}\left[\left(\tilde{X}_{k:}^T P^{-1}\beta^\star - y_k\right)^2\right] \leq (X_{k:}^T\beta^\star - y_k)^2 + \frac{1-p_m}{p_m}\beta^{\star T}\mathrm{diag}(X_{k:}X_{k:}^T)\beta^\star. \qquad (8)$$

As for the second term,

$$\mathbb{E}_{D_{k:}}\left[\|P^{-1}\mathrm{diag}(\tilde{X}_{k:}\tilde{X}_{k:}^T)\beta^\star\|^2\right] = \mathbb{E}_{D_{k:}}\left[\sum_{j=1}^d \tilde{X}_{kj}^4 p_j^{-2}\beta_j^{\star 2}\right]$$

$$= \sum_{j=1}^d X_{kj}^4 p_j^{-1}\beta_j^{\star 2}$$

$$\leq \frac{1}{p_m}\sum_{j=1}^d X_{kj}^4 \beta_j^{\star 2}$$

$$\leq \frac{1}{p_m}\left(\sum_{j=1}^d X_{kj}^2\right)\left(\sum_{j=1}^d X_{kj}^2\beta_j^{\star 2}\right)$$

$$= \frac{1}{p_m}\|X_{k:}\|^2\beta^{\star T}\mathrm{diag}(X_{k:}X_{k:}^T)\beta^\star$$

Finally, one obtains

$$\mathbb{E}[\|\tilde{g}_k(\beta^\star)\|^2 \mid \mathcal{F}_{k-1}] \leq \frac{1}{p_m^2}\mathbb{E}_{(X_{k:},y_k)}\left[(\epsilon_k)^2\|X_{k:}\|^2\right]$$
$$+ \frac{(1-p_m) + (1-p_m)^2}{p_m^3}\mathbb{E}_{(X_{k:},y_k)}\left[\|X_{k:}\|^2\beta^{\star T}\mathrm{diag}(X_{k:}X_{k:}^T)\beta^\star\right].$$

3 We aim at proving there exists $H$ such that

$$\mathbb{E}[\tilde{g}_k(\beta^\star)\tilde{g}_k(\beta^\star)^T] \preccurlyeq C = cH.$$

Simple computations lead to:

$$\mathbb{E}[\tilde{g}_k(\beta^\star)\tilde{g}_k(\beta^\star)^T] = \mathbb{E}[T_1 + T_2 + T_2^T + T_3],$$

with:

$$T_1 = (\tilde{X}_{k:}^T P^{-1}\beta^\star - y_k)^2 P^{-1}\tilde{X}_{k:}\tilde{X}_{k:}^T P^{-1},$$
$$T_2 = -(\tilde{X}_{k:}^T P^{-1}\beta^\star - y_k)P^{-1}\tilde{X}_{k:}\beta^{\star T}\mathrm{diag}(\tilde{X}_{k:}\tilde{X}_{k:}^T)P^{-2}(I - P),$$
$$T_3 = (I - P)P^{-2}\mathrm{diag}(\tilde{X}_{k:}\tilde{X}_{k:}^T)\beta^\star\beta^{\star T}\mathrm{diag}(\tilde{X}_{k:}\tilde{X}_{k:}^T)P^{-2}(I - P).$$

**Bound on $T_1$.** For the first term, we use

$$P^{-1}\tilde{X}_{k:}\tilde{X}_{k:}^T P^{-1} \preccurlyeq \frac{1}{p_m^2}\tilde{X}_{k:}\tilde{X}_{k:}^T, \tag{9}$$

since for all vector $v \neq 0$, $v^T\left(\frac{1}{p_m^2}\tilde{X}_{k:}\tilde{X}_{k:}^T - P^{-1}\tilde{X}_{k:}\tilde{X}_{k:}^T P^{-1}\right)v \geq 0$,

$$\sum_{j=1}^d \left(\frac{1}{p_m^2} - \frac{1}{p_j^2}\right)\tilde{X}_{kj}^2 v_j^2 + 2\sum_{1\leq j<l\leq d}\left(\frac{1}{p_m^2} - \frac{1}{p_j p_l}\right)\tilde{X}_{kj}\tilde{X}_{kl}v_j v_l$$

$$\overset{(iii)}{\geq} \sum_{j=1}^d \left(\frac{1}{p_m^2} - \frac{1}{p_j^2}\right)\tilde{X}_{kj}^2 v_j^2 + 2\sum_{1\leq j<l\leq d}\sqrt{\left(\frac{1}{p_m^2} - \frac{1}{p_j^2}\right)\left(\frac{1}{p_m^2} - \frac{1}{p_l^2}\right)}\tilde{X}_{kj}\tilde{X}_{kl}v_j v_l$$

$$= \left(\sum_{j=1}^d \sqrt{\left(\frac{1}{p_m^2} - \frac{1}{p_j^2}\right)}\tilde{X}_{kj}v_j\right)^2 \geq 0.$$

Step (iii) uses $\left(\frac{1}{p_m^2} - \frac{1}{p_j p_l}\right) \geq \sqrt{\left(\frac{1}{p_m^2} - \frac{1}{p_j^2}\right)\left(\frac{1}{p_m^2} - \frac{1}{p_l^2}\right)}$. Indeed,

$$\left(\frac{1}{p_m^2} - \frac{1}{p_j p_l}\right)^2 \geq \left(\frac{1}{p_m^2} - \frac{1}{p_j^2}\right)\left(\frac{1}{p_m^2} - \frac{1}{p_l^2}\right)$$

$$\Leftrightarrow \left(\frac{1}{p_m^4} - 2\frac{1}{p_j p_l}\frac{1}{p_m^2} + \frac{1}{p_j^2 p_l^2}\right) - \frac{1}{p_m^4} + \frac{1}{p_m^2 p_l^2} + \frac{1}{p_m^2 p_j^2} - \frac{1}{p_j^2 p_l^2} \geq 0$$

$$\Leftrightarrow \left(\frac{1}{p_m p_j} - \frac{1}{p_m p_l}\right)^2 \geq 0.$$

Let us now prove that

$$\frac{1}{p_m^2}\tilde{X}_{k:}\tilde{X}_{k:}^T \preccurlyeq \frac{1}{p_m^2}X_{k:}X_{k:}^T$$

i.e.

$$\tilde{X}_{k:}\tilde{X}_{k:}^T \preccurlyeq X_{k:}X_{k:}^T. \tag{10}$$

Indeed, for all vector $v \neq 0$, $v^T(X_{k:}X_{k:}^T - \tilde{X}_{k:}\tilde{X}_{k:}^T)v \geq 0$:

$$v^T(X_{k:}X_{k:}^T - \tilde{X}_{k:}\tilde{X}_{k:}^T)v = \sum_{j=1}^{d}(1 - \delta_{kj}^2)X_{kj}^2 v_j^2 + 2\sum_{1 \leq j < l \leq d}(1 - \delta_{kj}\delta_{kl})X_{kj}X_{kl}v_j v_l$$

$$\overset{(iv)}{\geq} \sum_{j=1}^{d}(1 - \delta_{kj}^2)X_{kj}^2 v_j^2 + 2\sum_{1 \leq j < l \leq d}\sqrt{(1 - \delta_{kj}^2)(1 - \delta_{kl}^2)}X_{kj}X_{kl}v_j v_l$$

$$= \left(\sum_{j=1}^{d}\sqrt{(1 - \delta_{kj}^2)}X_{kj}v_j\right)^2 \geq 0$$

Step (iv) is obtained using $(1 - \delta_{kj}\delta_{kl}) \geq \sqrt{(1 - \delta_{kj}^2)(1 - \delta_{kl}^2)}$. Indeed,

$$(1 - \delta_{kj}\delta_{kl})^2 \geq (1 - \delta_{kj}^2)(1 - \delta_{kl}^2) \Leftrightarrow (1 - 2\delta_{kl}\delta_{kj} + \delta_{kj}^2\delta_{kl}^2) - 1 + \delta_{kj}^2 - \delta_{kj}^2\delta_{kl}^2 + \delta_{kl}^2 \geq 0$$

$$\Leftrightarrow (\delta_{kj} - \delta_{kl})^2 \geq 0.$$

Then, by (8) and $(X_{k:}^T\beta^\star - y_k)^2 = \epsilon_k^2$,

$$\mathbb{E}_{(X_{k:},y_k)}[T_1] = \mathbb{E}_{(X_{k:},y_k)}\left[\frac{1}{p_m^2}\epsilon_k^2 X_{k:}X_{k:}^T\right] + \mathbb{E}_{(X_{k:},y_k)}\left[\frac{1 - p_m}{p_m^3}\left(\beta^{\star T}\mathrm{diag}(X_{k:}X_{k:}^T)\beta^\star\right)X_{k:}X_{k:}^T\right].$$

Noting that

$$\|\mathrm{diag}(X_{k:})\beta^\star\|^2 \leq \|X_{k:}\|^2\|\beta^\star\|^2, \tag{11}$$

$$\mathbb{E}[T_1] \preccurlyeq \frac{1}{p_m^2}\mathrm{Var}(\epsilon_k)H + \frac{1 - p_m}{p_m^3}\|X_{k:}\|^2\|\beta^\star\|^2 H \tag{12}$$

**Bound on $T_3$.** Using the resulting matrix structure of

$$(I - P)P^{-2}\mathrm{diag}(\tilde{X}_{k:}\tilde{X}_{k:}^T)\beta^\star\beta^{\star T}\mathrm{diag}(\tilde{X}_{k:}\tilde{X}_{k:}^T)P^{-2}(I - P),$$

detailed as follows

$$\begin{pmatrix} (\beta_1^\star)^2\delta_{k1}^4 X_{k1}^4 & \beta_1^\star\beta_2^\star\delta_{k1}^2\delta_{k2}^2 X_{k1}^2 X_{k2}^2 & \\ & \ddots & \\ & & (\beta_d^\star)^2\delta_{kd}^4 X_{kd}^4 \end{pmatrix},$$

one obtains

$$\mathbb{E}_{D_{k:}}[T_3] = \underbrace{(I-P)P^{-2}P\mathrm{diag}(X_{k:}X_{k:}{}^T)\beta^\star\beta^{\star T}\mathrm{diag}(X_{k:}X_{k:}{}^T)PP^{-2}(I-P)}_{=:T_{3a}}$$

$$+ \underbrace{(I-P)P^{-2}(P-P^2)\mathrm{diag}(X_{k:}X_{k:}{}^T)\mathrm{diag}(\beta^\star\beta^{\star T})\mathrm{diag}(X_{k:}X_{k:}{}^T)P^{-2}(I-P)}_{=:T_{3b}}. \quad (13)$$

Using similar arguments as in (9), both terms in (13) are bounded as follows

$$T_{3a} \preccurlyeq \frac{(1-p_m)^2}{p_m^2}\mathrm{diag}(X_{k:}X_{k:}{}^T)\beta^\star\beta^{\star T}\mathrm{diag}(X_{k:}X_{k:}{}^T)$$

$$T_{3b} \preccurlyeq \frac{(1-p_m)^3}{p_m^3}\mathrm{diag}(X_{k:}X_{k:}{}^T)\mathrm{diag}(\beta^\star\beta^{\star T})\mathrm{diag}(X_{k:}X_{k:}{}^T)$$

For $T_{3a}$, one can go further by using

$$\mathrm{diag}(X_{k:}X_{k:}^T)\beta^\star\beta^{\star T}\mathrm{diag}(X_{k:}X_{k:}^T) \preccurlyeq \|\mathrm{diag}(X_{k:})\beta^\star\|^2 X_{k:}X_{k:}^T. \quad (14)$$

Let us prove that for all vector $v \neq 0$,

$$v^T(\|\mathrm{diag}(X_{k:})\beta^\star\|^2 X_{k:}X_{k:}^T - \mathrm{diag}(X_{k:}X_{k:}^T)\beta^\star\beta^{\star T}\mathrm{diag}(X_{k:}X_{k:}^T))v \geq 0, \text{ i.e.}$$

$$\underbrace{\sum_{j=1}^d \left(\left(\sum_{l=1}^d X_{il}^2 \beta_{\star l}^2\right) X_{kj}^2 - X_{kj}^4 \beta_j^{\star 2}\right)v_j^2 + 2\sum_{1 \leq j < m \leq d}\left(\left(\sum_{l=1}^d X_{kl}^2 \beta_l^{\star 2}\right)X_{kj}X_{km} - \beta_j^\star \beta_m^\star X_{kj}^2 X_{km}^2\right)v_m v_j \geq 0}_{=:Q}$$

Indeed, $Q \geq \left(\sum_{j=1}^d \sqrt{\left(\sum_{l=1}^d X_{kl}^2 \beta_l^{\star 2}\right)X_{kj}^2 - X_{kj}^4\beta_j^{\star 2}}v_j\right)^2 \geq 0$, since, looking at the term depending only on $v_j v_m$:

$$\left(\left(\sum_{l=1}^d X_{kl}^2 \beta_l^{\star 2}\right)X_{kj}X_{km} - \beta_j^\star \beta_m^\star X_{kj}^2 X_{km}^2\right)$$

$$\geq \sqrt{\left(\left(\sum_{l=1}^d X_{kl}^2 \beta_l^{\star 2}\right)X_{kj}^2 - X_{kj}^4 \beta_j^{\star 2}\right)\left(\left(\sum_{l=1}^d X_{kl}^2 \beta_l^{\star 2}\right)X_{km}^2 - X_{km}^4 \beta_m^{\star 2}\right)}$$

is equivalent to

$$\left(\sum_{l=1}^d X_{kl}^2 \beta_l^{\star 2}\right)X_{kj}^4 X_{km}^2 \beta_j^{\star 2} + \left(\sum_{l=1}^d X_{kl}^2 \beta_l^{\star 2}\right)X_{km}^4 X_{kj}^2 \beta_m^{\star 2} - 2\left(\sum_{l=1}^d X_{kl}^2 \beta_l^{\star 2}\right)X_{kj}^3 X_{km}^3 \beta_j^\star \beta_m^\star \geq 0$$

$$\Leftrightarrow \left(\sqrt{\left(\sum_{l=1}^d X_{kl}^2 \beta_l^{\star 2}\right)}X_{kj}^2 X_{km}\beta_j^\star - \sqrt{\left(\sum_{l=1}^d X_{kl}^2 \beta_l^{\star 2}\right)}X_{km}^2 X_{kj}\beta_m^\star\right)^2 \geq 0$$

For $T_{3b}$, one can also dig deeper noting that

$$\mathrm{diag}(X_{k:}X_{k:}^T)\mathrm{diag}(\beta^\star\beta^{\star T})\mathrm{diag}(X_{k:}X_{k:}^T) \preccurlyeq \|\mathrm{diag}(X_{k:})\beta^\star\|^2 X_{k:}X_{k:}^T. \quad (15)$$

For all vector $v \neq 0$, we aim at proving

$$v^T(\|\beta^{\star T}\mathrm{diag}(X_{k:})\|^2 X_{k:}X_{k:}^T - \mathrm{diag}(X_{k:}X_{k:}^T)\mathrm{diag}(\beta^\star\beta^{\star T})\mathrm{diag}(X_{k:}X_{k:}^T)))v \geq 0$$

$$\Leftrightarrow \underbrace{\sum_{j=1}^d \left(\left(\sum_{l=1}^d X_{kl}^2 \beta_l^{\star 2}\right)X_{kj}^2 - X_{kj}^4 \beta_j^{\star 2}\right)v_j^2 + 2\sum_{1 \leq j < m \leq d}\left(\sum_{l=1}^d X_{kl}^2 \beta_l^{\star 2}\right)X_{kj}X_{km}v_j v_m \geq 0.}_{=:Q'}$$

Indeed, $Q' \geq \left( \sum_{j=1}^{d} \sqrt{\left( \sum_{l=1}^{d} X_{kl}^2 \beta_l^{\star 2} \right) X_{kj}^2 - X_{kj}^4 \beta_j^{\star 2}} v_j \right)^2 \geq 0$ since

$$\left( \left( \sum_{l=1}^{d} X_{kl}^2 \beta_l^{\star 2} \right) X_{kj} X_{km} \right)$$

$$\geq \sqrt{\left( \left( \sum_{l=1}^{d} X_{kl}^2 \beta_l^{\star 2} \right) X_{kj}^2 - X_{kj}^4 \beta_j^{\star 2} \right) \left( \left( \sum_{l=1}^{d} X_{kl}^2 \beta_l^{\star 2} \right) X_{km}^2 - X_{km}^4 \beta_m^{\star 2} \right)}$$

$$\iff \left( \sum_{l=1}^{d} X_{kl}^2 \beta_l^{\star 2} \right) X_{kj}^4 X_{km}^2 \beta_j^{\star 2} + \left( \sum_{l=1}^{d} X_{kl}^2 \beta_l^{\star 2} \right) X_{km}^4 X_{kj}^2 \beta_m^{\star 2} - X_{kj}^4 X_{km}^4 \beta_j^{\star 2} \beta_m^{\star 2} \geq 0$$

Combining (11), (14) and (15) lead to

$$\mathbb{E}_{(X_{k:}, y_k)} [T_{3a}] \preccurlyeq \frac{(1 - p_m)^2}{p_m^2} \|X_{k:}\|^2 \|\beta^\star\|^2 H$$

$$\mathbb{E}_{(X_{k:}, y_k)} [T_{3b}] \preccurlyeq \frac{(1 - p_m)^3}{p_m^3} \|X_{k:}\|^2 \|\beta^\star\|^2 H$$

and to the final bound for $T_3$,

$$\mathbb{E} [T_3] \preccurlyeq \frac{(1 - p_m)^2}{p_m^2} \|X_{k:}\|^2 \|\beta^\star\|^2 H + \frac{(1 - p_m)^3}{p_m^3} \|X_{k:}\|^2 \|\beta^\star\|^2 H. \tag{16}$$

**Bound on $T_2 + T_2^T$.** Firstly, focus on $T_2$:

$$T_2 = -(\tilde{X}_{k:}^T P^{-1} \beta^\star - y_k) P^{-1} \tilde{X}_{k:} \beta^{\star T} \text{diag}(\tilde{X}_{k:} \tilde{X}_{k:}^T) P^{-2} (I - P)$$
$$=: -(A - B),$$

where

$$A = P^{-1} \tilde{X}_{k:} \tilde{X}_{k:}^T P^{-1} \beta^\star \beta^{\star T} \text{diag}(\tilde{X}_{k:} \tilde{X}_{k:}^T) P^{-2} (I - P)$$
$$B = P^{-1} \tilde{X}_{k:} y_k \beta^{\star T} \text{diag}(\tilde{X}_{k:} \tilde{X}_{k:}^T) P^{-2} (I - P).$$

**Computation w.r.t. $\mathbb{E}_{D_{k:}}$.** Term $A$ can be split into three terms, denoting $\tilde{\mathbb{X}}_k := \tilde{X}_{k:} \tilde{X}_{k:}^T$

$$A_1 = P^{-1} \text{diag}(\tilde{\mathbb{X}}_k) P^{-1} \beta^\star \beta^{\star T} \text{diag}(\tilde{\mathbb{X}}_k) P^{-2} (I - P)$$
$$A_2 = P^{-1} (\tilde{\mathbb{X}}_k - \text{diag}(\tilde{\mathbb{X}}_k)) P^{-1} \text{diag}(\beta^\star \beta^{\star T}) \text{diag}(\tilde{\mathbb{X}}_k) P^{-2} (I - P)$$
$$A_3 = P^{-1} (\tilde{\mathbb{X}}_k - \text{diag}(\tilde{\mathbb{X}}_k)) P^{-1} (\beta^\star \beta^{\star T} - \text{diag}(\beta^\star \beta^{\star T})) \text{diag}(\tilde{\mathbb{X}}_k) P^{-2} (I - P).$$

Noting that
$$A_1 = P^{-2} \text{diag}(\tilde{X}_{k:} \tilde{X}_{k:}^T) \beta^\star \beta^{\star T} \text{diag}(\tilde{X}_{k:} \tilde{X}_{k:}^T) P^{-2} (I - P),$$
the expectation $\mathbb{E}_{D_{k:}}$ has already been computed in (13), so

$$\mathbb{E}_{D_{k:}} [A_1] = P^{-2} P \text{diag}(X_{k:} X_{k:}^T) \beta^\star \beta^{\star T} \text{diag}(X_{k:} X_{k:}^T) P P^{-2} (I - P)$$
$$+ P^{-2} (P - P^2) \text{diag}(X_{k:} X_{k:}^T) \text{diag}(\beta^\star \beta^{\star T}) \text{diag}(X_{k:} X_{k:}^T) P^{-2} (I - P). \tag{17}$$

As for $A_2$, making the structure of $P^{-1} (\tilde{X}_{k:} \tilde{X}_{k:}^T - \text{diag}(\tilde{X}_{k:} \tilde{X}_{k:}^T)) P^{-1} \text{diag}(\beta^\star \beta^{\star T}) \text{diag}(\tilde{X}_{k:} \tilde{X}_{k:})$ explicit,

$$A_2 = \begin{pmatrix} 0 & \frac{1}{p_1 p_2} \delta_{k1} \delta_{k2}^3 X_{k1} X_{k2}^3 \beta_2^{\star 2} & \cdots & \frac{1}{p_1 p_d} \delta_{k1} \delta_{kd}^3 X_{k1} X_{kd}^3 \beta_d^{\star 2} \\ \frac{1}{p_1 p_2} \delta_{k2} \delta_{k1}^3 X_{k2} X_{k1}^3 \beta_1^{\star 2} & 0 & & \\ & & \ddots & \\ \frac{1}{p_1 p_d} \delta_{kd} \delta_{k1}^3 X_{kd} X_{k1}^3 \beta_1^{\star 2} & & & 0 \end{pmatrix},$$

one has

$$\mathbb{E}_{D_{k:}}[A_2] = (X_{k:}X_{k:}{}^T - \mathrm{diag}(X_{k:}X_{k:}{}^T))\mathrm{diag}(\beta^\star\beta^{\star T})\mathrm{diag}(X_{k:}X_{k:}{}^T)P^{-2}(I-P). \quad (18)$$

As for $A_3$, the term $P^{-1}(\tilde{X}_{k:}\tilde{X}_{k:}^T - \mathrm{diag}(\tilde{X}_{k:}\tilde{X}_{k:}^T))P^{-1}(\beta^\star\beta^{\star T} - \mathrm{diag}(\beta^\star\beta^{\star T}))\mathrm{diag}(\tilde{X}_{k:}\tilde{X}_{k:})$ can be made explicit as

$$
\begin{pmatrix}
\sum_{l=2}^{d} \frac{1}{p_1 p_l}\delta_{kl}X_{kl}\beta_l^\star \delta_{k1}X_{k1}^3\beta_1^\star & \sum_{l=3}^{d}\frac{1}{p_1 p_l}\delta_{kl}X_{kl}\beta_l^\star \delta_{k1}\delta_{k2}^2 X_{k1}X_{k2}^2\beta_2^\star & \cdots & \sum_{l\neq 1,d}\frac{1}{p_1 p_l}\delta_{kl}X_{kl}\beta_l^\star \delta_{k1}\delta_{kd}^2 X_{k1}X_{kd}^2\beta_d^\star \\
& \ddots & & \\
& & \ddots & \\
& & & \sum_{l=1}^{d-1}\frac{1}{p_l p_d}\delta_{kl}X_{kl}\beta_l^\star \delta_{kd}X_{kd}^3\beta_d^\star
\end{pmatrix}
$$

which gives

$$
\begin{aligned}
\mathbb{E}_{D_{k:}}[A_3] &= (X_{k:}X_{k:}^T - \mathrm{diag}(X_{k:}X_{k:}^T))(\beta^\star\beta^{\star T} - \mathrm{diag}(\beta^\star\beta^{\star T}))\mathrm{diag}(X_{k:}X_{k:}^T)PP^{-2}(I-P) \\
&\quad + (I-P)\mathrm{diag}\left((X_{k:}X_{k:}^T - \mathrm{diag}(X_{k:}X_{k:}^T))(\beta^\star\beta^{\star T} - \mathrm{diag}(\beta^\star\beta^{\star T}))\mathrm{diag}(X_{k:}X_{k:}^T)\right)P^{-2}(I-P).
\end{aligned}
$$

Noting the following,

$$
\begin{aligned}
&\mathrm{diag}\left((\tilde{X}_{k:}\tilde{X}_{k:}^T - \mathrm{diag}(\tilde{X}_{k:}\tilde{X}_{k:}^T))(\beta^\star\beta^{\star T} - \mathrm{diag}(\beta^\star\beta^{\star T}))\mathrm{diag}(\tilde{X}_{k:}\tilde{X}_{k:})\right) \\
&\qquad = \mathrm{diag}\left(\tilde{X}_{k:}\tilde{X}_{k:}^T\beta^\star\beta^{\star T}\mathrm{diag}(\tilde{X}_{k:}\tilde{X}_{k:})\right) - \mathrm{diag}(\tilde{X}_{k:}\tilde{X}_{k:}^T)\mathrm{diag}(\beta^\star\beta^{\star T})\mathrm{diag}(\tilde{X}_{k:}\tilde{X}_{k:}),
\end{aligned}
$$

one has

$$
\begin{aligned}
\mathbb{E}_{D_{k:}}&[A_3] \\
&= P^{-1}P(\tilde{X}_{k:}\tilde{X}_{k:}^T - \mathrm{diag}(\tilde{X}_{k:}\tilde{X}_{k:}^T))(\beta^\star\beta^{\star T} - \mathrm{diag}(\beta^\star\beta^{\star T}))\mathrm{diag}(\tilde{X}_{k:}\tilde{X}_{k:}^T)PP^{-2}(I-P) \\
&\quad + P^{-1}(P-P^2)\mathrm{diag}\left(\tilde{X}_{k:}\tilde{X}_{k:}^T\beta^\star\beta^{\star T}\mathrm{diag}(\tilde{X}_{k:}\tilde{X}_{k:}^T)\right)P^{-2}(I-P) \\
&\quad - P^{-1}(P-P^2)\mathrm{diag}(\tilde{X}_{k:}\tilde{X}_{k:}^T)\mathrm{diag}(\beta^\star\beta^{\star T})\mathrm{diag}(\tilde{X}_{k:}\tilde{X}_{k:}^T)P^{-2}(I-P) \quad (19)
\end{aligned}
$$

Term $B$ can be made explicit as follows

$$
P^{-1}\tilde{X}_{k:}\beta^{\star T}\mathrm{diag}(\tilde{X}_{k:}\tilde{X}_{k:}) = \begin{pmatrix}
\frac{1}{p_1}\beta_1^\star\delta_{i1}^3 X_{i1}^3 & \frac{1}{p_1}\beta_1^\star\delta_{i1}^2\delta_{i2}X_{i1}^2 X_{i2} & \\
\frac{1}{p_2}\beta_2^\star\delta_{i2}^2 X_{i2}^2\delta_{i1}X_{i1} & \frac{1}{p_2}\beta_2^\star\delta_{i2}^3 X_{i2}^3 & \\
& & \ddots
\end{pmatrix}
$$

which implies

$$
\begin{aligned}
\mathbb{E}_{D_{k:}}[B] &= y_k X_{k:}\beta^{\star T}\mathrm{diag}(X_{k:}X_{k:}{}^T)PP^{-2}(I-P) \\
&\qquad + y_k(I-P)\mathrm{diag}(X_{k:}\beta^{\star T}\mathrm{diag}(X_{k:}X_{k:}{}^T))P^{-2}(I-P). \quad (20)
\end{aligned}
$$

Putting Equations (17), (18), (19) and (20) together,

$$
\mathbb{E}\left[T_2 + T_2^T\right] = \mathbb{E}_{(X_{k:},y_k)}\left[T_{21} + T_{22} + T_{23} + T_{23}^T + T_{24} + T_{24}^T + T_{25}\right]
$$

$$
\begin{aligned}
T_{21} &= -2(P^{-1} - I)\mathrm{diag}(X_{k:}X_{k:}{}^T)\beta^\star\beta^{\star T}\mathrm{diag}(X_{k:}X_{k:}{}^T)(P^{-1} - I) \\
T_{22} &= -2P^{-3}((I-P)(I - 3P + 2P^2)\mathrm{diag}(X_{k:}X_{k:}{}^T)\mathrm{diag}(\beta^\star\beta^{\star T})\mathrm{diag}(X_{k:}X_{k:}{}^T) \\
T_{23} &= -X_{k:}X_{k:}{}^T\mathrm{diag}(\beta^\star\beta^{\star T})\mathrm{diag}(X_{k:}X_{k:}{}^T)(P^{-2}(I-P) - P^{-1}(I-P)) \\
T_{24} &= -(X_{k:}{}^T\beta^\star - y_k)X_{k:}\beta^{\star T}\mathrm{diag}(X_{k:}X_{k:}{}^T)P^{-1}(I-P) \\
T_{25} &= -2(X_{k:}{}^T\beta^\star - y_k)(I-P)\mathrm{diag}(X_{k:}\beta^{\star T}\mathrm{diag}(X_{k:}X_{k:}{}^T))P^{-2}(I-P),
\end{aligned}
$$

**Computation w.r.t.** $\mathbb{E}_{(X_{k:},y_k)}$.   For $T_{21}$, it trivially holds that

$$-\mathrm{diag}(X_{k:}X_{k:}^T)\beta^\star\beta^{\star T}\mathrm{diag}(X_{k:}X_{k:}^T) \preccurlyeq 0. \tag{21}$$

Indeed, for all vector $v \neq 0$,

$$\sum_{j=1}^{d} X_{kj}^4 \beta_j^{\star 2} v_j^2 + 2 \sum_{1 \leq j < m \leq d} \beta_j^\star \beta_m^\star X_{kj}^2 X_{km}^2 v_j v_m = \left( \sum_{j=1}^{d} X_{kj}^2 \beta_j^\star v_j \right)^2 \geq 0.$$

Denoting the maximum of the coefficients of $P$ as $p_M = \max_j p_j$, one has

$$T_{21} \preccurlyeq -2 \frac{(1-p_M)^2}{p_m^2} \mathrm{diag}(X_{k:}X_{k:}{}^T)\beta^\star\beta^{\star T}\mathrm{diag}(X_{k:}X_{k:}{}^T)$$

$$\preccurlyeq 0 \qquad\qquad\qquad\qquad\qquad\qquad \text{(using (21))}.$$

$T_{22}$ is split into two terms,

$$T_{22a} = -2P^{-3}((I-P)(I+2P^2))\mathrm{diag}(X_{k:}X_{k:}{}^T)\mathrm{diag}(\beta^\star\beta^{\star T})\mathrm{diag}(X_{k:}X_{k:}{}^T)$$
$$T_{22b} = 6P^{-2}(I-P)\mathrm{diag}(X_{k:}X_{k:}{}^T)\mathrm{diag}(\beta^\star\beta^{\star T})\mathrm{diag}(X_{k:}X_{k:}{}^T)$$

$$T_{22a} \preccurlyeq -2 \frac{(1-p_M)(1+2p_M^2)}{p_m^3} \mathrm{diag}(X_{k:}X_{k:}{}^T)\mathrm{diag}(\beta^\star\beta^{\star T})\mathrm{diag}(X_{k:}X_{k:}{}^T) \preccurlyeq 0,$$

since it is a diagonal matrix with only negative coefficients, and noting that $\frac{(1-p_M)(1+2p_M^2)}{p_m^3} > 0$.
Then,

$$T_{22b} \preccurlyeq \frac{6(1-p_m)}{p_m^2} \mathrm{diag}(X_{k:}X_{k:}{}^T)\mathrm{diag}(\beta^\star\beta^{\star T})\mathrm{diag}(X_{k:}X_{k:}{}^T)$$

which implies

$$\mathbb{E}_{(X_{k:},y_k)}[T_{22b}] \preccurlyeq \frac{6(1-p_m)}{p_m^2} \|X_{k:}\|^2 \|\beta^\star\|^2 H$$

using (14) and (11).

As for $T_{23} + T_{23}^T$, note that

$$T_{23} + T_{23}^T \preccurlyeq -2 \frac{(p_M-1)^2}{p_m^2} \big( X_{k:}X_{k:}{}^T \mathrm{diag}(\beta^\star\beta^{\star T})\mathrm{diag}(X_{k:}X_{k:}{}^T)$$
$$+ \mathrm{diag}(\beta^\star\beta^{\star T})\mathrm{diag}(X_{k:}X_{k:}{}^T)X_{k:}X_{k:}{}^T \big)$$

One prove that

$$- \big( X_{k:}X_{k:}{}^T \mathrm{diag}(\beta^\star\beta^{\star T})\mathrm{diag}(X_{k:}X_{k:}{}^T) + \mathrm{diag}(X_{k:}X_{k:}{}^T)\mathrm{diag}(\beta^\star\beta^{\star T})X_{k:}X_{k:}{}^T \big)$$
$$\preccurlyeq -2 \left( \min_{j=1,\dots,d} \beta_j^{\star 2} X_{kj}^2 \right) X_{k:}X_{k:}^T \quad (22)$$

Indeed, denoting $m = \left( \min_{j=1,\dots,d} \beta_j^{\star 2} X_{kj}^2 \right)$, one has

$$v^T\Big(-2mX_{k:}X_{k:}^T+(X_{k:}X_{k:}{}^T\mathrm{diag}(\beta^\star\beta^{\star T})\mathrm{diag}(X_{k:}X_{k:}{}^T)$$
$$+\mathrm{diag}(X_{k:}X_{k:}{}^T)\mathrm{diag}(\beta^\star\beta^{\star T})X_{k:}X_{k:}{}^T)\Big)v\geq 0$$

$$\Leftrightarrow\sum_{j=1}^{d}\left(-2mX_{kj}^2+2X_{kj}^4\beta_j^{\star 2}\right)v_j^2$$
$$+2\sum_{1\leq j<q\leq d}\left(-2mX_{kj}X_{kq}+X_{kj}^3X_{kq}\beta_j^{\star 2}+X_{kq}^3X_{kj}\beta_q^{\star 2}\right)v_jv_q\geq 0$$

$$\Leftrightarrow\sum_{j=1}^{d}\left(-2mX_{kj}^2+2X_{kj}^4\beta_j^{\star 2}\right)v_j^2$$
$$+2\sum_{1\leq j<q\leq d}\sqrt{\left(-2mX_{kj}^2+2X_{kj}^4\beta_j^{\star 2}\right)\left(-2mX_{kq}^2+2X_{kq}^4\beta_q^{\star 2}\right)}v_jv_q\geq 0$$

$$\Leftrightarrow\left(\sum_{j=1}^{d}\sqrt{\left(-2mX_{kj}^2+2X_{kj}^4\beta_j^{\star 2}\right)}v_j\right)^2\geq 0,$$

using that

$$\left(-2mX_{kj}^2+2X_{kj}^4\beta_j^{\star 2}\right)\left(-2mX_{kq}^2+2X_{kq}^4\beta_q^{\star 2}\right)$$
$$\leq\left(-2mX_{kj}X_{kq}+X_{kj}^3X_{kq}\beta_j^{\star 2}+X_{kq}^3X_{kj}\beta_q^{\star 2}\right)^2$$
$$\Leftrightarrow\left(X_{kj}^3X_{kq}\beta_j^{\star 2}-X_{kq}^3X_{kj}\beta_q^{\star 2}\right)^2\geq 0$$

Therefore

$$\mathbb{E}_{(X_{k:},y_k)}\left[T_{23}+T_{23}^T\right]\preccurlyeq-2\frac{(p_M-1)^2}{p_m^2}\left(\min_{j=1,\dots,d}\beta_j^{\star 2}X_{kj}^2\right)H\preccurlyeq 0,$$

since $H$ is definite positive.

Finally one uses $(X_{k:}^T\beta^\star-y_k)=\epsilon_k$ to conclude by independence that $T_{24}=T_{25}=0$.

One gets

$$\mathbb{E}\left[T_2+T_2^T\right]\preccurlyeq\frac{6(1-p_m)}{p_m^2}\|X_{k:}\|^2\|\beta^\star\|^2H. \tag{23}$$

Combining (12), (16) and (23) leads to the desired bound.

$\square$

## S2.3  Proof of Lemma 3

**Lemma 6.** *For all $k\geq 0$, given the binary mask $D$, the adjusted gradient $\tilde{g}_k(\beta)$ is a.s. $L_{k,D}$-Lipschitz continuous, i.e. for all $u,v\in\mathbb{R}^d$,*

$$\|\tilde{g}_k(u)-\tilde{g}_k(v)\|\leq L_{k,D}\|u-v\|\ a.s..$$

*Set*

$$L:=\sup_{k,D}L_{k,D}\leq\frac{1}{p_m^2}\max_k\|X_{k:}\|^2\ a.s..$$

*In addition, for all $k\geq 0$, $\tilde{g}_k(\beta)$ is almost surely co-coercive.*

*Proof.* Note that

$$\|\tilde{g}_k(u) - \tilde{g}_k(v)\| = \left\| \left( P^{-1}\tilde{X}_{k:}\tilde{X}_{k:}^T P^{-1} - (I-P)P^{-2}\mathrm{diag}(\tilde{X}_{k:}\tilde{X}_{k:}^T) \right)(u-v) \right\|$$

$$\leq \left\| \left( P^{-1}\tilde{X}_{k:}\tilde{X}_{k:}^T P^{-1} - (I-P)P^{-2}\mathrm{diag}(\tilde{X}_{k:}\tilde{X}_{k:}^T) \right) \right\| \|u-v\|$$

$$\leq \left\| \frac{1}{p_m^2} \left( \tilde{X}_{k:}\tilde{X}_{k:}^T - (1-p_m)\mathrm{diag}(\tilde{X}_{k:}\tilde{X}_{k:}^T) \right) \right\| \|u-v\|$$

$$\leq \frac{1}{p_m^2} \|\tilde{X}_{k:}\|^2 \|u-v\|,$$

where we have used the Weyl inequality in the last step.

One can thus choose $L_{k,D} = \frac{1}{p_m^2}\|\tilde{X}_{k:}\|^2$ and

$$L = \sup_{k,D} L_{k,D} \leq \frac{1}{p_m^2} \sup_k \|X_{k:}\|^2 \leq \frac{1}{p_m^2} \max_k \|X_{k:}\|^2$$

Then, let us prove that the primitive of the adjusted gradient $\tilde{g}_k$ is convex. To do this, we check that the derivative of $\tilde{g}_k$ is definite positive:

$$\frac{\partial}{\partial \beta}\tilde{g}_k(\beta) = \frac{1}{p^2} \left( \tilde{X}_{k:}\tilde{X}_{k:}^T - (1-p)\mathrm{diag}\left( \tilde{X}_{k:}\tilde{X}_{k:}^T \right) \right)$$

since $\left( \tilde{X}_{k:}\tilde{X}_{k:}^T - (1-p)\mathrm{diag}\left( \tilde{X}_{k:}\tilde{X}_{k:}^T \right) \right)$ is positive semi-definite. Indeed,

$$v^T \left( \tilde{X}_{k:}\tilde{X}_{k:}^T - (1-p)\mathrm{diag}\left( \tilde{X}_{k:}\tilde{X}_{k:}^T \right) \right) v \geq 0$$

$$\Leftrightarrow \sum_{j=1}^{d} p\tilde{X}_{kj}^2 v_j^2 + 2 \sum_{1\leq j<l\leq d} \tilde{X}_{kj}\tilde{X}_{kl}v_j v_l \geq 0$$

$$\Leftrightarrow \left( \sum_{j=1}^{d} \sqrt{p}\tilde{X}_{kj}v_j \right)^2 \geq 0,$$

using $p^2 \left( \tilde{X}_{kj} \right)^2 \left( \tilde{X}_{kj} \right)^2 \leq \left( \tilde{X}_{kj} \right)^2 \left( \tilde{X}_{kl} \right)^2$ since $p \leq 1$. $\qquad\square$

## S3 Proof of the theoretical convergence rate with estimated missing probabilities $(\hat{p}_j)_j$

In this section, we consider that we access $2n$ observations $(\tilde{X}'_{k:}, y'_k)_{1\leq k\leq n}$ and $(\tilde{X}_{k:}, y_k)_{1\leq k\leq n}$: we want to control the error of the estimator built with our Algorithm 1 using the second $n$ observations with the proportions $\hat{p}$ *estimated* using the first $n$ observations. In practice, it is likely that estimating the proportions on the same points used for running the algorithm would not hurt the performance. However, the proof requires the estimation of $p$ and the stochastic gradient to be independent, we thus have to split the dataset. As we aim at proving that the convergence speed remains of $O(1/n)$, the induced multiplicative factor 2 on $n$ will not modify the order of the convergence rate.

More precisely, we consider the proportions estimated on the points $(\tilde{X}'_{k:}, y'_k)_{1\leq k\leq n}$, for $1 \leq j \leq d$:

$$\hat{p}_j = \frac{1}{n} \sum_{k=1}^{n} \mathbb{1}_{X'_{kj}\neq \mathrm{NA}}. \tag{24}$$

Moreover, in the exceptional case that $\hat{p}_j = 0$ (which would correspond to a feature that is *never present* in the first half of the dataset, and thus would probably be discarded in practice), we correct the estimated proportion to $n^{-1}$. That is $\hat{p}_j = \max(n^{-1}, \frac{1}{n}\sum_{k=1}^{n} \mathbb{1}_{X'_{kj}\neq \mathrm{NA}})$. We do so only to ensure that $\hat{p}_j > 0$ which is necessary in the algorithm.

We then build the sequence $(\hat{\beta}_k)_{k\geq 0}$ of iterates constructed with an estimated value of the missing probabilities $\hat{p} = (\hat{p}_j)_{1\leq j\leq d} \in \mathbb{R}^d$ as follows:

$$\begin{cases} \hat{\beta}_0 = \beta_0 \\ \hat{\beta}_k = h_k(\hat{\beta}_{k-1}, \hat{p}) := \hat{\beta}_{k-1} - \alpha \tilde{g}_k(\hat{\beta}_{k-1}, \hat{p}), \ k > 0 \end{cases} \tag{25}$$

where

$$\tilde{g}_k(\beta_k, \hat{p}) := \hat{P}^{-1}\tilde{X}_{k:}\left(\tilde{X}_{k:}^T\hat{P}^{-1}\beta_k - y_k\right) - (I - \hat{P})\hat{P}^{-2}\text{diag}\left(\tilde{X}_{k:}\tilde{X}_{k:}^T\right)\beta_k$$

with $\hat{P} = \text{diag}((\hat{p}_j)_{1\leq j\leq d})$.

We denote the averaged iterates of $(\hat{\beta}_k)_{k\geq 0}$ by $(\bar{\hat{\beta}}_k)_{k\geq 0}$ such that $\bar{\hat{\beta}}_k = \frac{1}{k+1}\sum_{\ell=0}^k \hat{\beta}_\ell$.

**Theorem 2** (Convergence rate with estimated missing probabilities). *Assume that the missing probabilities $(\hat{p}_j)_{j=1,\ldots,d}$ are estimated as in Equation (24) using $(\tilde{X}'_{k:}, y'_k)_{1\leq k\leq n}$ and $\hat{\beta}_k$ given in Equation (25) is constructed using $(\tilde{X}_{k:}, y_k)_{1\leq k\leq n}$.*

*There exists an event $A_n = \{\forall j \in \{1,\ldots,d\}, \hat{p}_j > p_j/2\}$ with high probability $\mathbb{P}(A_n) \geq 1 - de^{-np_m/8}$. For any constant step-size $\alpha \leq \frac{1}{2L}$, Algorithm 1 ensures that, for any $n \geq 1$,*

$$\mathbb{E}\left[\|\bar{\hat{\beta}}_n - \beta_\star\|_{H^{1/2}}^2 \big| A_n\right] \leq 2\underbrace{\mathbb{E}\left[\|\bar{\beta}_n - \beta_\star\|_{H^{1/2}}^2\right]}_{\text{Bounded by Theorem 1}} + \underbrace{\frac{2^6}{p_m^6}\frac{1}{\gamma\mu}CL\frac{5d}{n} + \frac{2^8}{p_m^6}\frac{1}{\gamma\mu}CL\frac{d(1-p_m)^{2n}}{n^2}}_{\text{Residual term due to the estimation of }\hat{p}},$$

*where $p_m = \min_{j=1,\ldots d} p_j$, $\gamma = \alpha\left(1 - \frac{\alpha L}{2}\right)$ and $C = \left(1 + \frac{1}{\gamma\mu}\right)\alpha^2 dC_{\text{obs}}$, where $L$ is given in Equation (6) and $C_{\text{obs}}$ is such that $\mathbb{E}[\|\tilde{X}_{k:}\|^4(|\tilde{X}_{k:}^T\hat{\beta}_{k-1}| + |y_k|)^4] \leq C_{\text{obs}}^2, \forall k \geq 0$, and where we denote $\|v\|_{H^{1/2}}^2 = \|H^{1/2}v\|_2^2$ and $\mu$ the smallest eigenvalue of $H = \mathbb{E}_{(X_{k:},y_k)}[X_{k:}X_{k:}^T]$ which is assumed to be positive.*

*In addition, one has the following unconditional result, for any $n \geq 1$,*

$$R(\bar{\hat{\beta}}_n) - R(\beta_\star) = \frac{1}{2}\mathbb{E}\left[\|\bar{\hat{\beta}}_n - \beta_\star\|_{H^{1/2}}^2\right]$$

$$\leq \underbrace{\mathbb{E}\left[\|\bar{\beta}_n - \beta_\star\|_{H^{1/2}}^2\right]}_{\text{Bounded by Theorem 1}} + \underbrace{\frac{1}{\gamma\mu}CL\left(\frac{2^4}{p_m^6}\frac{5d}{n} + \frac{2^6}{p_m^6}\frac{d(1-p_m)^{2n}}{n^2} + n^6\sqrt{d}e^{-np_m/16}\right)}_{\text{Residual term due to the estimation of }\hat{p}}.$$

*Proof.* The probability of $A_n$ is given by Lemma 8. Let us first remark

$$\mathbb{E}\left[\|\bar{\hat{\beta}}_n - \beta^\star\|_{H^{1/2}}^2\right] = \mathbb{E}\left[\|\bar{\hat{\beta}}_n - \bar{\beta}_n + \bar{\beta}_n - \beta_\star\|_{H^{1/2}}^2\right]$$

$$\leq 2\left(\mathbb{E}\left[\|\bar{\hat{\beta}}_n - \bar{\beta}_n\|_{H^{1/2}}^2\right] + \mathbb{E}\left[\|\bar{\beta}_n - \beta_\star\|_{H^{1/2}}^2\right]\right).$$

Note that for the first expectation in the last inequality, the randomness comes from the estimated proportions $(\hat{p}_j)_j$ and from the samples $(\tilde{X}_{k:}, y_k)_{1\leq k\leq n}$, whereas for the second expectation, the randomness is only due to $(\tilde{X}_{k:}, y_k)_{1\leq k\leq n}$. We then combine Theorem 1 and Lemma 9:

- Theorem 1 (and more precisely Remark 3) gives the bound for $\mathbb{E}\left[\|\bar{\beta}_n - \beta_\star\|_{H^{1/2}}^2\right]$. Note that for the conditional result, $\mathbb{E}\left[\|\bar{\beta}_n - \beta_\star\|_{H^{1/2}}^2 | A_n\right] = \mathbb{E}\left[\|\bar{\beta}_n - \beta_\star\|_{H^{1/2}}^2\right]$, because $\bar{\beta}_n \perp\!\!\!\perp A_n$.

- One has

$$\mathbb{E}\left[\|\bar{\hat{\beta}}_n - \bar{\beta}_n\|_{H^{1/2}}^2\right] \leq L\mathbb{E}\left[\|\bar{\hat{\beta}}_n - \bar{\beta}_n\|^2\right]$$

$$= L\mathbb{E}\left[\|(n+1)^{-1}\sum_{k=0}^n \hat{\beta}_k - \beta_k\|^2\right]$$

$$\leq L(n+1)^{-1}\sum_{k=0}^n \mathbb{E}\left[\|\hat{\beta}_k - \beta_k\|^2\right]$$

The result follows by using Lemma 9.

$\square$

We first prove the following key Lemma, that will be the main element in the proof of Lemma 9.

**Lemma 7.** *For all $k \geq 0$, one has*

$$\mathbb{E}\left[\|\hat{\beta}_k - \beta_k\|^2\right] \leq \frac{1}{\gamma\mu} C \left(\mathbb{E}\left[\frac{\|\hat{p} - p\|^4}{\min_{j=1,\ldots,d}(p_j, \hat{p}_j)^{12}}\Big|\hat{p}\right]\right)^{1/2},$$

*with $\gamma = \alpha\left(1 - \frac{\alpha L}{2}\right)$ and $C = \left(1 + \frac{1}{\gamma\mu}\right)\alpha^2 d C_{\text{obs}}$, where $L$ is given in Equation (6) and $C_{\text{obs}}$ is such that $\left(\mathbb{E}[\|\tilde{X}_{k:}\|^4(|\tilde{X}_{k:}^T\hat{\beta}_{k-1}| + |y_k|)^4]\right)^{1/2} \leq C_{\text{obs}}$, for all $k \geq 0$.*

*Proof.* Let us denote $\delta_k^2 := \|\hat{\beta}_k - \beta_k\|^2 = \|h_k(\hat{\beta}_{k-1}, \hat{p}) - h_k(\beta_{k-1}, p)\|^2$. We first remark that

$$\mathbb{E}[\delta_k^2] = \mathbb{E}[\mathbb{E}[\delta_k^2|\hat{p}]],$$

so that we bound the conditional expectation $\mathbb{E}[\delta_k^2|\hat{p}]$. In the following, to control the deviation of $\hat{\beta}_k$ to $\beta_k$, we use

1. the deviation resulting from the use of $\hat{p}$ instead of $p$ to construct $\hat{\beta}_k$ (term 1 in (26)),

2. the deviation resulting from the use of $\hat{\beta}_{k-1}$ as a support point instead of $\beta_{k-1}$ (term 2 in (26)).

To do so, we introduce a "ghost" sequence (never computed) $h_k(\hat{\beta}_{k-1}, p)$. Noting that for any $\eta > 0$, and any $a, b \in \mathbb{R}^d$, we have $\|a + b\|^2 \leq (1 + \eta)\|a\|^2 + (1 + \eta^{-1})\|b\|^2$, we have:

$$\mathbb{E}[\delta_k^2|\hat{p}] = \mathbb{E}\left[\|h_k(\hat{\beta}_{k-1}, \hat{p}) - h_k(\hat{\beta}_{k-1}, p) + h_k(\hat{\beta}_{k-1}, p) - h_k(\beta_{k-1}, p)\|^2|\hat{p}\right]$$

$$\leq (1 + \frac{1}{\eta})\underbrace{\mathbb{E}\left[\|h_k(\hat{\beta}_{k-1}, \hat{p}) - h_k(\hat{\beta}_{k-1}, p)\|^2|\hat{p}\right]}_{\text{term 1}} + (1 + \eta)\underbrace{\mathbb{E}\left[\|h_k(\hat{\beta}_{k-1}, p) - h_k(\beta_{k-1}, p)\|^2|\hat{p}\right]}_{\text{term 2}}.$$

$$(26)$$

We control both terms separately.

**Control of term 1.** Almost surely, we have the following

$$\|h_k(\hat{\beta}_{k-1}, \hat{p}) - h_k(\hat{\beta}_{k-1}, p)\|^2 = \|\hat{\beta}_{k-1} - \alpha\tilde{g}_k(\hat{\beta}_{k-1}, \hat{p}) - \hat{\beta}_{k-1} + \alpha\tilde{g}_k(\hat{\beta}_{k-1}, p)\|^2$$

$$= \alpha^2\|\tilde{g}_k(\hat{\beta}_{k-1}, p) - \tilde{g}_k(\hat{\beta}_{k-1}, \hat{p})\|^2$$

$$= \alpha^2\sum_{j=1}^{d}(\tilde{g}_{kj}(\hat{\beta}_{k-1}, p) - \tilde{g}_{kj}(\hat{\beta}_{k-1}, \hat{p}))^2,$$

where $\tilde{g}_{kj}(\hat{\beta}_{k-1}, p)$ denotes the $j$-th component of the vector $\tilde{g}_k(\hat{\beta}_{k-1}, p) \in \mathbb{R}^d$. We introduce the function $\psi_{kj} : [0, 1] \to \mathbb{R}$ such that $\psi_{kj}(t) = \tilde{g}_{kj}(\hat{\beta}_{k-1}, p + t(\hat{p} - p))$. By the mean value theorem, one has

$$\|h_k(\hat{\beta}_{k-1}, \hat{p}) - h_k(\hat{\beta}_{k-1}, p)\|^2 = \alpha^2\sum_{j=1}^{d}(\psi_{kj}(1) - \psi_{kj}(0))^2$$

$$\leq \alpha^2\sum_{j=1}^{d}\sup_{t\in[0,1]}(\psi_{kj}'(t))^2 \qquad (27)$$

Yet, $\psi_{kj}'(t) = \left\langle\nabla\tilde{g}_{kj}(\hat{\beta}_{k-1}, p + t(\hat{p} - p)), \hat{p} - p\right\rangle$. Using the Cauchy–Schwarz inequality and denoting $p_t = p + t(\hat{p} - p)$, one obtains

$$\alpha^2\sum_{i=1}^{d}\sup_{t\in[0,1]}(\psi_{kj}'(t))^2 \leq \alpha^2\sum_{j=1}^{d}\sup_{t\in[0,1]}\|\nabla\tilde{g}_{kj}(\hat{\beta}_{k-1}, p_t)\|^2\|\hat{p} - p\|^2 \qquad (28)$$

Recall that $p_t = ((p_t)_1, \ldots, (p_t)_d)^T \in \mathbb{R}^d$. Using the form of the debiased gradient given in Remark 2, one has for $1 \leq j \leq d$:

$$\tilde{g}_{kj}(\hat{\beta}_{k-1}, p_t) = \underbrace{\left( \left( \begin{array}{ccccc} \frac{1}{(p_t)_1 (p_t)_j} & \cdots & \underbrace{\frac{1}{(p_t)_j}}_{j\text{th position}} & \cdots & \frac{1}{(p_t)_d (p_t)_j} \end{array} \right) \odot \tilde{X}_{kj} \tilde{X}_{k:}^T \right) \hat{\beta}_{k-1}}_{\text{denoted } \tilde{g}_{kj}^1(\hat{\beta}_{k-1}, p_t)}$$

$$+ \underbrace{\left( \begin{array}{ccc} \frac{1}{(p_t)_1} & \cdots & \frac{1}{(p_t)_d} \end{array} \right) \odot \tilde{X}_{kj} y_k}_{\text{denoted } \tilde{g}_{kj}^2(\hat{\beta}_{k-1}, p_t)}$$

One has $\|\nabla \tilde{g}_{kj}(\hat{\beta}_{k-1}, p_t)\|^2 = \sum_{\ell=1}^d \left( \frac{\partial \tilde{g}_{kj}}{\partial ((p_t)_\ell)}(\hat{\beta}_{k-1}, p_t) \right)^2$ with

$$\frac{\partial \tilde{g}_{kj}^1}{\partial ((p_t)_\ell)}(\hat{\beta}_{k-1}, p_t) = \begin{cases} \left( \frac{-1}{(p_t)_j^2} \left( \begin{array}{ccccc} \frac{1}{(p_t)_1} & \cdots & \underbrace{1}_{j\text{th position}} & \cdots & \frac{1}{(p_t)_d} \end{array} \right) \odot \tilde{X}_{kj} \tilde{X}_{k:}^T \right) \hat{\beta}_{k-1} & \text{if } \ell = j \\[3em] \left( \frac{1}{(p_t)_j} \left( \begin{array}{ccccc} 0 & \cdots & \underbrace{\frac{-1}{(p_t)_\ell^2}}_{\ell\text{th position}} & \cdots & 0 \end{array} \right) \odot \tilde{X}_{kj} \tilde{X}_{k:}^T \right) \hat{\beta}_{k-1} & \text{otherwise} \end{cases}$$

and

$$\frac{\partial \tilde{g}_{kj}^2}{\partial ((p_t)_\ell)}(\hat{\beta}_{k-1}, p_t) = \left( \begin{array}{ccccc} 0 & \cdots & \underbrace{\frac{-1}{(p_t)_\ell^2}}_{\ell\text{th position}} & \cdots & 0 \end{array} \right) \odot \tilde{X}_{kj} y_k$$

Therefore, $\forall \ell \in \{0, \ldots, d\}$,

$$\left( \frac{\partial \tilde{g}_{kj}}{\partial ((p_t)_\ell)}(\hat{\beta}_{k-1}, p_t) \right)^2 \leq \frac{1}{(p_t)_{\min}^6} (\tilde{X}_{kj}(|\tilde{X}_{k:}^T \hat{\beta}_{k-1}| + |y_k|))^2$$

with $(p_t)_{\min} = \min_{j=1,\ldots,d} (p_t)_j$ which leads to

$$\|\nabla \tilde{g}_{kj}(\hat{\beta}_{k-1}, p_t)\|^2 = \sum_{\ell=1}^d \left( \frac{\partial \tilde{g}_{kj}}{\partial ((p_t)_\ell)}(\hat{\beta}_{k-1}, p_t) \right)^2 \leq \frac{1}{(p_t)_{\min}^6} \|\tilde{X}_{k:}\|^2 (|\tilde{X}_{k:}^T \hat{\beta}_{k-1}| + |y_k|)^2.$$

One obtains, plugging the equation above into Equations (27) and (28):

$$\|h_k(\hat{\beta}_{k-1}, \hat{p}) - h_k(\hat{\beta}_{k-1}, p)\|^2 \leq \alpha^2 d \sup_{t \in [0,1]} \frac{1}{(p_t)_{\min}^6} \|\tilde{X}_{k:}\|^2 (|\tilde{X}_{k:}^T \hat{\beta}_{k-1}| + |y_k|)^2 \|\hat{p} - p\|^2$$

and finally, taking expectation conditionally to $\hat{p}$:

$$\mathbb{E}[\|h_k(\hat{\beta}_{k-1}, \hat{p}) - h_k(\hat{\beta}_{k-1}, p)\|^2 | \hat{p}] \leq \alpha^2 d \mathbb{E} \left[ \|\tilde{X}_{k:}\|^2 (\tilde{X}_{k:}^T \hat{\beta}_{k-1} + y_k)^2 \sup_{t \in [0,1]} \frac{\|\hat{p} - p\|^2}{(p_t)_{\min}^6} \Big| \hat{p} \right]$$

Assuming that $\left( \mathbb{E}[\|\tilde{X}_{k:}\|^4 (|\tilde{X}_{k:}^T \hat{\beta}_{k-1}| + |y_k|)^4] \right)^{1/2} \leq C_{\text{obs}}$, one has, by Cauchy Schwartz,

$$\mathbb{E}[\|h_k(\hat{\beta}_{k-1}, \hat{p}) - h_k(\hat{\beta}_{k-1}, p)\|^2 | \hat{p}] \leq \alpha^2 d C_{\text{obs}} \mathbb{E}^{1/2} \left[ \sup_{t \in [0,1]} \frac{\|\hat{p} - p\|^4}{(p_t)_{\min}^{12}} \Big| \hat{p} \right]$$

$$\leq \alpha^2 d C_{\text{obs}} \mathbb{E}^{1/2} \left[ \frac{\|\hat{p} - p\|^4}{\min_{j=1,\ldots,d}(p_j, \hat{p}_j)^{12}} \Big| \hat{p} \right]. \qquad (29)$$

**Control of term 2.** We now control the part of the distance coming form the fact that the true-iterate $h_k(\beta_{k-1}, p)$ and ghost-iterate $h_k(\hat{\beta}_{k-1}, p)$ updates are computed at two different points $\hat{\beta}_{k-1}$ and $\beta_{k-1}$:

$$\|h_k(\hat{\beta}_{k-1}, p) - h_k(\beta_{k-1}, p)\|^2 = \|\hat{\beta}_{k-1} - \beta_{k-1} - \alpha(\tilde{g}_k(\hat{\beta}_{k-1}, \hat{p}) - \tilde{g}_k(\beta_{k-1}, p))\|^2$$

$$\leq \|\hat{\beta}_{k-1} - \beta_{k-1}\|^2 - 2\alpha \left\langle \hat{\beta}_{k-1} - \beta_{k-1}, \tilde{g}_k(\hat{\beta}_{k-1}, \hat{p}) - \tilde{g}_k(\beta_{k-1}, p) \right\rangle$$

$$+ \alpha^2 \|\tilde{g}_k(\hat{\beta}_{k-1}, \hat{p}) - \tilde{g}_k(\beta_{k-1}, p)\|^2$$

Using Lemma 3 which gives the co-coercivity of the debiased gradient, one obtains

$$\alpha^2 \|\tilde{g}_k(\hat{\beta}_{k-1}, p) - \tilde{g}_k(\beta_{k-1}, p)\|^2 \leq \alpha^2 L \left\langle \hat{\beta}_{k-1} - \beta_{k-1}, \tilde{g}_k(\hat{\beta}_{k-1}, p) - \tilde{g}_k(\beta_{k-1}, p) \right\rangle.$$

It implies that

$$\|h_k(\hat{\beta}_{k-1}, p) - h_k(\beta_{k-1}, p)\|^2 \leq \|\hat{\beta}_{k-1} - \beta_{k-1}\|^2$$

$$- 2\alpha \left(1 - \frac{\alpha L}{2}\right) \left\langle \hat{\beta}_{k-1} - \beta_{k-1}, \tilde{g}_k(\hat{\beta}_{k-1}, p) - \tilde{g}_k(\beta_{k-1}, p) \right\rangle$$

Denoting $\gamma = \alpha\left(1 - \frac{\alpha L}{2}\right)$, one has

$$\mathbb{E}\left[\|h_k(\hat{\beta}_{k-1}, p) - h_k(\beta_{k-1}, p)\|^2|\hat{p}\right] \leq \mathbb{E}\left[\|\hat{\beta}_{k-1} - \beta_{k-1}\|^2|\hat{p}\right]$$

$$- 2\gamma \mathbb{E}\left[\left\langle \hat{\beta}_{k-1} - \beta_{k-1}, \tilde{g}_k(\hat{\beta}_{k-1}, p) - \tilde{g}_k(\beta_{k-1}, p) \right\rangle|\hat{p}\right]$$

$$\leq \mathbb{E}\left[\|\hat{\beta}_{k-1} - \beta_{k-1}\|^2|\hat{p}\right]$$

$$- 2\gamma \left\langle \mathbb{E}\left[\hat{\beta}_{k-1} - \beta_{k-1}\right], \nabla R(\hat{\beta}_{k-1}) - \nabla R(\beta_{k-1}) \right\rangle,$$

using that $\mathbb{E}[\tilde{g}_k(\hat{\beta}_{k-1}, p)|\hat{p}] = \nabla R(\hat{\beta}_{k-1})$ because $\hat{\beta}_{k-1}$ is constructed with $n$ observations independent of the ones using for computing $\hat{p}$ (it implies $\tilde{g}_k(\cdot, p) \perp\!\!\!\perp \hat{p}$).

Using the $\mu$-strong convexity of $R$, one has.

$$\mathbb{E}\left[\|h_k(\hat{\beta}_{k-1}, p) - h_k(\beta_{k-1}, p)\|^2|\hat{p}\right] \leq (1 - 2\gamma\mu)\mathbb{E}\left[\|\hat{\beta}_{k-1} - \beta_{k-1}\|^2|\hat{p}\right] \qquad (30)$$

**Conclusion.** Choosing $\eta = \gamma\mu$ in Equation (26) with Equation (29) and Equation (30) leads to:

$$\mathbb{E}[\delta_k^2] \leq \left(1 + \frac{1}{\gamma\mu}\right) \alpha^2 d C_{\text{obs}} \mathbb{E}^{1/2}\left[\frac{\|\hat{p} - p\|^4}{\min_{j=1,\ldots,d}(p_j, \hat{p}_j)^{12}}\Big|\hat{p}\right] + (1 - \gamma\mu)\mathbb{E}[\delta_{k-1}^2|\hat{p}].$$

Denoting $C := \left(1 + \frac{1}{\gamma\mu}\right) \alpha^2 d C_{\text{obs}}$,

$$\mathbb{E}[\delta_{k+1}^2|\hat{p}] \leq (1 - \gamma\mu)^k \mathbb{E}[\delta_0^2|\hat{p}] + C \sum_{i=0}^{k} (1 - \gamma\mu)^{k-i} \mathbb{E}^{1/2}\left[\frac{\|\hat{p} - p\|^4}{\min_{j=1,\ldots,d}(p_j, \hat{p}_j)^{12}}\Big|\hat{p}\right]$$

$$= (1 - \gamma\mu)^k \mathbb{E}[\delta_0^2] + C \frac{1 - (1 - \gamma\mu)^k}{\gamma\mu} \mathbb{E}^{1/2}\left[\frac{\|\hat{p} - p\|^4}{\min_{j=1,\ldots,d}(p_j, \hat{p}_j)^{12}}\Big|\hat{p}\right]$$

$$\leq \frac{1}{\gamma\mu} C \mathbb{E}^{1/2}\left[\frac{\|\hat{p} - p\|^4}{\min_{j=1,\ldots,d}(p_j, \hat{p}_j)^{12}}\Big|\hat{p}\right], \qquad (31)$$

where in the last inequality we used that $\mathbb{E}[\delta_0^2|\hat{p}] = 0$.

$\square$

**Lemma 8.** *Let* $A_n = \{\forall j \in \{1, \ldots, d\}, \hat{p}_j > p_j/2\}$ *be the event where the missing probabilities are not under-estimated by a factor of at least two. The probability of this event is such that*

$$\mathbb{P}(A_n) \geq 1 - de^{-np_m/8},$$

*where* $p_m = \min_{j=1,\ldots d} p_j$.

*Proof.* We use the multiplicative Chernoff-Hoeffding inequality: if $X_1, \ldots, X_n$ are i.i.d. variables such that $\mathbb{E}[\sum_{i=1}^{n} X_i] = \mathbb{E}[X] = \mu$, one has

$$\mathbb{P}(X \leq (1-\delta)\mu) \leq e^{-\delta^2 \mu/2}, \; 0 \leq \delta \leq 1.$$

Fix $j \in \{1, \ldots, d\}$. Choosing $\delta = 1/2$ and applying the Chernoff-Hoeffding inequality to $n\hat{p}_j = \sum_{i=1}^{n} \delta_{ij}$ with $\delta_{ij} \overset{\text{i.i.d.}}{\sim} \mathcal{B}(p_j)$, one has

$$\mathbb{P}(n\hat{p}_j \leq np_j/2) \leq e^{-np_j/8}$$

implying that $\mathbb{P}(\hat{p}_j \leq p_j/2) \leq e^{-np_j/8}$.

Finally,

$$\begin{aligned}
\mathbb{P}(A_n) &= 1 - \mathbb{P}(\exists j \in \{1, \ldots, d\}, \hat{p}_j \leq p_j/2) \\
&\geq 1 - \sum_{j=1}^{d} \mathbb{P}(\hat{p}_j \leq p_j/2) \\
&\geq 1 - \sum_{j=1}^{d} e^{-np_j/8} \geq 1 - de^{-np_m/8}.
\end{aligned}$$

$\square$

**Lemma 9.** *Let $A_n = \{\forall j \in \{1, \ldots, d\}, \hat{p}_j > p_j/2\}$. For any $k \geq 0$,*

$$\mathbb{E}\left[\|\hat{\beta}_k - \beta_k\|^2 | A_n\right] \leq \frac{2^5}{p_m^6} \frac{1}{\gamma\mu} C \frac{d}{n} + \frac{2^7}{p_m^6} \frac{1}{\gamma\mu} C \frac{d(1-p_m)^{2n}}{n^2},$$

*with $p_m = \min_{j=1,\ldots d} p_j$, $\gamma = \alpha\left(1 - \frac{\alpha L}{2}\right)$ and $C = \left(1 + \frac{1}{\gamma\mu}\right)\alpha^2 d C_{\text{obs}}$, where $L$ is given in Equation* (6) *and $C_{\text{obs}}$ is such that $\left(\mathbb{E}[\|\tilde{X}_{k:}\|^4(|\tilde{X}_{k:}^T \hat{\beta}_{k-1}| + |y_k|)^4]\right)^{1/2} \leq C_{\text{obs}}, \forall k \geq 0$.*

*In addition, $\forall k \geq 0$,*

$$\mathbb{E}\left[\|\hat{\beta}_k - \beta_k\|^2\right] \leq \frac{1}{\gamma\mu} C \left(\frac{2^4}{p_m^6} \frac{5d}{n} + \frac{2^6}{p_m^6} \frac{d(1-p_m)^{2n}}{n^2} + n^6\sqrt{d}e^{-np_m/16}\right).$$

*Proof.* We start by proving the result conditional to the event $A_n$. We recall that

$$\mathbb{E}[\|\hat{\beta}_k - \beta_k\|^2 | A_n] = \frac{\mathbb{E}[\|\hat{\beta}_k - \beta_k\|^2 \mathbb{1}_{A_n}]}{\mathbb{P}(A_n)}$$

Using Lemma 8, one has $\mathbb{P}(A_n) \geq 1/2$. Thus,

$$\mathbb{E}[\|\hat{\beta}_k - \beta_k\|^2 | A_n] \leq 2\mathbb{E}[\|\hat{\beta}_k - \beta_k\|^2 \mathbb{1}_{A_n}].$$

By Lemma 7, it leads to

$$\mathbb{E}[\|\hat{\beta}_k - \beta_k\|^2 | A_n] \leq \frac{2}{\gamma\mu} C \left(\mathbb{E}\left[\frac{\|\hat{p} - p\|^4}{\min_{j=1,\ldots,d}(p_j, \hat{p}_j)^{12}}\Big|\hat{p}\right]\right)^{1/2}$$

Yet, almost surely $\frac{\mathbb{1}_{A_n}}{\min_{j=1,\ldots,d}(p_j, \hat{p}_j)^6} \leq \frac{2^6}{p_m^6}$ given that on the event $A_n$, $\forall j = 1, \ldots, d$, $\hat{p}_j > p_j/2$. We thus have

$$\mathbb{E}[\|\hat{\beta}_k - \beta_k\|^2 | A_n] \leq \frac{2^7}{p_m^6} \frac{1}{\gamma\mu} C \mathbb{E}^{1/2}\left[\|\hat{p} - p\|^4\right]$$

Moreover, $\mathbb{E}\left[\|\hat{p} - p\|^4\right] = \mathbb{E}(\sum_{j=1}^{d}(\hat{p}_j - p_j)^2)^2 = (\sum_{j=1}^{d} \mathbb{E}(\hat{p}_j - p_j)^4) + \sum_{j,j'=1,j\neq j'}^{d}(\mathbb{E}(\hat{p}_j - p_j)^2)(\mathbb{E}(\hat{p}_{j'} - p_{j'})^2)$, by independence of the estimation of each coordinate.

We thus have to compute the 4-th order moment and the quadratic error of $\hat{p}_j$.

First, $\mathbb{E}(\hat{p}_j - p_j)^2 = \text{Var}(\hat{p}_j) + \text{Bias}(\hat{p}_j)^2$, with $\text{Var}(\hat{p}_j) = \text{Var}(\frac{1}{n}\sum_{k=1}^{n} p_j(1-p_j)) = \frac{1}{n}(p_j(1-p_j)) \leq \frac{1}{4n}$.

By denoting $\hat{p}_j^{nc} = \frac{1}{n}\sum_{k=1}^{n} \mathbb{1}_{X'_{kj} \neq \text{NA}}$, one has $\mathbb{E}[\hat{p}_j^{nc}] = p_j$ and thus

$$\text{Bias}(\hat{p}_j) = \mathbb{E}[\hat{p}_j - p_j] = \mathbb{E}[\hat{p}_j - \hat{p}_j^{nc}] = \mathbb{E}\left[\frac{1}{n}\mathbb{1}_{\hat{p}_j^{nc} \neq 0}\right] = \frac{1}{n}(1-p_j)^n \leq \frac{1}{n}(1-p_m)^n.$$

Thus $\sum_{j,j'=1, j \neq j'}^{d}(\mathbb{E}(\hat{p}_j - p_j)^2)(\mathbb{E}(\hat{p}_{j'} - p_{j'})^2) \leq d^2\left(\frac{1}{4n} + \left(\frac{1}{n}(1-p_m)^n\right)^2\right)^2$.

On the other hand, for the 4-th order moment, $\mathbb{E}((\hat{p}_j - p_j)^4) = \frac{1}{n^4}(n\mu_{4,p_j} + 3n(n-1)p_j^2(1-p_j)^2)$, with $\mu_{4,p_j} = p_j(1-p_j)^4 + p_j^4(1-p_j) \leq 1/12$ (this is the classical computation of the 4-th moment of a binomial random variable). Overall $\sum_{j=1}^{d}\mathbb{E}((\hat{p}_j - p_j)^4) \leq d(\frac{1}{12n^3} + \frac{3}{16n^2}) \leq \frac{d}{n^2} \leq \frac{d^2}{n^2}$.

Combining the second order and fourth order terms , $\mathbb{E}\left[\|\hat{p} - p\|^4\right] \leq \frac{d^2}{n^2} + d^2\left(\frac{5}{4n} + \left(\frac{1}{n}(1-p_m)^n\right)^2\right)^2 \leq d^2\left(\frac{5}{4n} + \left(\frac{1}{n}(1-p_m)^n\right)^2\right)^2$.

Therefore,

$$(\mathbb{E}\left[\|\hat{p} - p\|^4\right])^{1/2} \leq d\left(\frac{5}{4n} + \left(\frac{1}{n}(1-p_m)^n\right)^2\right) \tag{32}$$

which implies

$$\mathbb{E}[\|\hat{\beta}_k - \beta_k\|^2 | A_n] \leq \frac{2^5}{p_m^6}\frac{5}{\gamma\mu}C\frac{d}{n} + \frac{2^7}{p_m^6}\frac{1}{\gamma\mu}C\frac{d(1-p_m)^{2n}}{n^2}.$$

For the unconditional result, one splits the term to control on the event $A_n$ and $A_n^c$, i.e.

$$\mathbb{E}^{\frac{1}{2}}\left[\frac{\|\hat{p} - p\|^4}{\min_{j=1,\ldots,d}(p_j,\hat{p}_j)^{12}}\right] \leq \mathbb{E}^{\frac{1}{2}}\left[\frac{\|\hat{p} - p\|^4\mathbb{1}_{A_n}}{\min_{j=1,\ldots,d}(p_j,\hat{p}_j)^{12}}\right] + \mathbb{E}^{\frac{1}{2}}\left[\frac{\|\hat{p} - p\|^4\mathbb{1}_{A_n^c}}{\min_{j=1,\ldots,d}(p_j,\hat{p}_j)^{12}}\right]. \tag{33}$$

On the event $A_n$, one has $\hat{p}_j \geq p_j/2$ which leads to $\frac{1}{\min_{j=1,\ldots,d}(p_j,\hat{p}_j)^6} \leq \frac{2^6}{p_j^6} \leq \frac{2^6}{p_m^6}$. Using Equation (32), one has

$$\mathbb{E}^{1/2}\left[\frac{\|\hat{p} - p\|^4}{\min_{j=1,\ldots,d}(p_j,\hat{p}_j)^{12}}\mathbb{1}_{A_n}\right] \leq \frac{2^4}{p_m^6}\frac{5d}{n} + \frac{2^6}{p_m^6}\frac{d(1-p_m)^{2n}}{n^2}. \tag{34}$$

On the event $A_n^c$, one has $\hat{p}_j \leq p_j/2 \leq p_j$.

$$\mathbb{E}^{1/2}\left[\frac{\|\hat{p} - p\|^4}{\min_{j=1,\ldots,d}(p_j,\hat{p}_j)^{12}}\mathbb{1}_{A_n^c}\right] \leq \mathbb{E}^{1/2}\left[\frac{1}{\hat{p}_j^{12}}\mathbb{1}_{A_n^c}\right]$$

As we have chosen to assign the minimal value $n^{-1}$ to $\hat{p}_j$ in the rare event that the empirical proportion was 0, we have the lower bound $\hat{p}_j \geq \frac{1}{n}$ which implies

$$\mathbb{E}^{1/2}\left[\frac{\|\hat{p} - p\|^4}{\min_{j=1,\ldots,d}(p_j,\hat{p}_j)^{12}}\mathbb{1}_{A_n^c}\right] \leq \mathbb{E}^{1/2}\left[\frac{1}{\hat{p}_j^{12}}\mathbb{1}_{A_n^c}\right] \leq \sqrt{n^{12}\mathbb{P}(A_n^c)} = n^6\sqrt{d}e^{-np_m/16}, \tag{35}$$

using Lemma 8. Combining Equations (33) to (35), one obtains

$$\mathbb{E}^{1/2}\left[\frac{\|\hat{p} - p\|^4}{\min_{j=1,\ldots,d}(p_j,\hat{p}_j)^{12}}\right] \leq \left(\frac{2^4}{p_m^6}\frac{5d}{n} + \frac{2^6}{p_m^6}\frac{d(1-p_m)^{2n}}{n^2} + n^6\sqrt{d}e^{-np_m/16}\right)$$

Finally, with the bound given in Lemma 7, one has

$$\mathbb{E}[\|\hat{\beta}_k - \beta_k\|^2] \leq \frac{1}{\gamma\mu}C\left(\frac{2^4}{p_m^6}\frac{5d}{n} + \frac{2^6}{p_m^6}\frac{d(1-p_m)^{2n}}{n^2} + n^6\sqrt{d}e^{-np_m/16}\right).$$

$\square$

Extension of such a result to cases without strong convexity (or independently of $\mu$) is an interesting open direction.

## S4  Add-on to Section 5: Lipschitz constant computation

The Lipschitz constant $L$ given in (6) is either computed from the complete covariates (oracle estimate) $\hat{L}_n^{\text{OR}} = \frac{1}{\hat{p}_m^2} \max_{1 \leq k \leq n} \|X_{k:}\|^2$, or estimated from the incomplete data matrix, $\hat{L}_n^{\text{NA}} = \frac{1}{\hat{p}_m^2} \max_{1 \leq k \leq n} \frac{\|\tilde{X}_{k:}\|^2 d}{\sum_j D_{kj}}$, with $\hat{p}_m = \min_{1 \leq j \leq d} \hat{p}_j$, and $\hat{p}_j = \frac{\sum_k D_{kj}}{n}$. In $\hat{L}_n^{\text{NA}}$, the squared norm of each row $\|\tilde{X}_{k:}\|^2$ is divided by the proportion of observed values $\frac{d}{\sum_j D_{kj}}$. This way, the value of $\|\tilde{X}_{k:}\|^2$ is renormalized, by taking into account that some rows may contain more missing values than others. Note that theoretically the step size has to satisfy $\alpha \leq \frac{1}{2\hat{L}_n^{\text{NA}}}$, thus $\hat{L}_n^{\text{NA}}$ may be overestimated but should not be underestimated at the risk of instability in Algorithm 1. Figure S7 shows that using a slightly overestimated Lipschitz constant estimate does not deteriorate the convergence obtained using the oracle estimate.

Figure S7: Empirical excess risk $(R_n(\beta_k) - R_n(\beta^\star))$ given $n$ for synthetic data ($n = 10^5$, $d = 10$) when there is 30% MCAR data, with 1 pass over the data and estimating the Lipschitz constant.

## S5  Add-on to Section 5: Handling polynomial missing features

The debiased averaged SGD algorithm proposed in Section 3 can be further extended to the case of polynomial features by using a different debiasing than in Equation (4).

For example, in dimension $d = 2$, with second-order polynomial features, the interaction effect of $X_{k1}X_{k2}$ and the effects of $X_{k1}^2$, $X_{k2}^2$ are accounted, so the augmented matrix design can be written as

$$(X_{:1}|X_{:2}|X_{:1}X_{:2}|X_{:1}^2|X_{:2}^2)^T.$$

Then, the "descent" direction at iteration $k$ in Equation (4) should be chosen as

$$U^{\odot -1} \odot \tilde{X}_{k:}\tilde{X}_{k:}^T \beta_k - \text{diag}(U)^{\odot -1} \odot \tilde{X}_{k:}y_k.$$

where

$$U = \begin{pmatrix} p_1 & p_1 p_2 & p_1 p_2 & p_1 & p_1 p_2 \\ p_1 p_2 & p_2 & p_1 p_2 & p_1 p_2 & p_2 \\ p_1 p_2 & p_1 p_2 & p_1 p_2 & p_1 p_2 & p_1 p_2 \\ p_1 & p_1 p_2 & p_1 p_2 & p_1 & p_1 p_2 \\ p_1 p_2 & p_2 & p_1 p_2 & p_1 p_2 & p_2 \end{pmatrix},$$

and $\text{diag}(U)$ denotes the vector formed by the diagonal coefficients of $U$ and $U^{\odot -1}$ stands for the matrix formed of the inverse coefficients of $U$.

**Synthetic data.** Considering a second-order model, we simulate data according to $y = (\tilde{X}_{:1}X_{:2}|X_{:1}^2|X_{:2}^2)^T\beta^\star + \epsilon$. An additional experiment is given in Figure S8 in Appendix S4, illustrating that Algorithm 1 still achieves a rate of $\mathcal{O}\left(\frac{1}{n}\right)$ while dealing with polynomial features of degree 2.

Figure S8: Empirical excess risk $(R_n(\beta_k) - R_n(\beta^\star))$ given $n$ for synthetic data ($n = 10^5$, $d = 10$) when the model accounts mixed effects.

**Real dataset.** About large-scale setting there is no computational barrier to apply the proposed method in high dimension, as the computational cost is similar to standard SGD strategies without missing data. These are computationally cheap at each iteration and particularly relevant on large datasets. In this section, we propose to run the proposed algorithm on the superconductivity dataset as in Subsection 5.3. $30\%$ of missing values are uniformly introduced in the initial 81 features, with $n = 21263$. However, here we consider polynomial features of order 2, which increases the initial dimension 81 to 3400.

The empirical proportions of missing values for each variable in the resulting dataset are represented on Figure S9, and the observed convergence rate for one pass on the data is displayed in Figure S10. With the same numerical complexity, Algorithm 1 performs as well as an averaged SGD strategy run on the complete observations, whereas a standard SGD strategy run on imputed-by-0 data saturates far from the optimum.

Figure S9: Proportion of missing values for the polynomial features of degree 2 on the superconductivity dataset, when the initial missingness proportion on the raw features is $30\%$.

Figure S10: Empirical excess risk $(R_n(\beta_k) - R_n(\beta^\star))$ given $n$ for the superconductivity dataset ($n = 21263$) (containing 81 initial features) and $d = 3403$ with polynomial features of degree 2. Three different algorithms are compared: an averaged SGD on complete data (blue), the proposed debiased averaged SGD Algorithm 1 (orange) and an averaged SGD run on imputed-by-0 data without any debiasing (green).

## S6  Add-on to Section 5: Description of the TraumaBase data variables

The variables of the TraumaBase dataset which are used in experiments are the following:

- *Lactate*: The conjugate base of lactic acid.
- *Delta.Hemo*: The difference between the homoglobin on arrival at hospital and that in the ambulance.
- *VE*: A volume expander is a type of intravenous therapy that has the function of providing volume for the circulatory system.
- *RBC*: A binary index which indicates whether the transfusion of Red Blood Cells Concentrates is performed.
- *SI*: Shock index indicates level of occult shock based on heart rate (*HR*) and systolic blood pressure (*SBP*). $SI = \frac{HR}{SBP}$. Evaluated on arrival at hospital.
- *HR*: Heart rate measured on arrival of hospital.
- *Age*: Age.