[Reviews · NeurIPS 2020]

Review 1

Summary and Contributions: This paper proposes an averaged stochastic gradient algorithm to handle missing data in linear regressions, and proves the convergence rate of O(1/n) in both streaming and finite-sample settings.

Strengths: The paper is sound. It provides theoretical guarantees as well as comprehensive numerical results. The proposed algorithm is efficient and achieves the optimal convergence rate.

Weaknesses: It would improve the quality of the paper if authors can provide a more in-depth discussion of the bounded feature assumption. The discussion after Thm 4 seems quite optimistic. If it is the case, it might be good to collect such results in an appendix. The notations are not clearly introduced. For example, in Algorithm 1, the first diag in Line 3 and the second diag in equation (4) seem to have different definitions.

Correctness: I believe them to be correct.

Clarity: The paper is well written.

Relation to Prior Work: The paper gives a nice review of and comparison with existing works.

Reproducibility: Yes

Additional Feedback:


Review 2

Summary and Contributions: The paper proposes an averaged stochastic gradient algorithm handling missing values in linear models. In oposite to the most models in this filed, this approach has the merit to be free from the need of any data distribution modeling. The authors consider a linear regression model to test the effectiveness of this approach.

Strengths: The idea of the paper is interesting and the methodology is justified theoretically.

Weaknesses: 1.The paper is hard to follow. I recommand text correction. Some notations appear before their definition, e.g. in line 137 'c(\Beta)' or the authors refer to Table 3, which does not exist but there is Figure 3. 2. On synthetic data, where missing rate was 30%, the authors compare the proposed approach with only one paper https://arxiv.org/pdf/1702.07098.pdf. I suggest the authors to perform an additional experiment on more benchmark datasets with 25%, 50%, 75% missing rates and comparing with more state-of-the-art methods.

Correctness: The theory is rather correct but it is hard to follow. I have discussed most of my concerns above.

Clarity: The paper is hard to follow. Text needs to be cleaned up.

Relation to Prior Work: Yes

Reproducibility: Yes

Additional Feedback: Post rebuttal ============================================== Following the authors response and in seeing the concerns raised by the other reviewers. I do not have a strong opinion and still consider it as borderline article although the authors managed to answer most of my concerns in an additional experiment.


Review 3

Summary and Contributions: This paper proposed a method that combines inverse probability weighting (IPW) and stochastic gradient descent (SGD) that leads to an optimization algorithm applicable to missing data.

Strengths: All the proposed methods are reasonable and the derivation is very clear. The problem being considered is an important problem in machine learning and statistics.

Weaknesses: The main weakness is perhaps the setup is too simple, at least in the missing data part; see additional feedbacks for more details (item 4). Literatures on the idea of IPW are missing; see additional feedbacks for more details (item 2).

Correctness: The derivations seem to be correct and the numerical results are reasonable.

Clarity: Yes, I have no difficulty reading this paper.

Relation to Prior Work: Yes the paper discussed prior work on the case without missing data.

Reproducibility: Yes

Additional Feedback: Comments. 1. A key quantity in the analysis is p_j, the probability of missing j-th feature. Is this quantity given in advanced? Under MCAR, it can be easily estimated. But some clarifications are needed. 2. Inverse probability weighting (IPW). Essentially, this method is just SGD + IPW (though it is coordinate-wise IPW not the regular IPW). The IPW is a very common approach for handling missing data, see, e.g., Chapter 3 of the following book > Little, R. J., & Rubin, D. B. (2019). Statistical analysis with missing data (Vol. 793). John Wiley & Sons. and the following review paper: > Seaman, S. R., & White, I. R. (2013). Review of inverse probability weighting for dealing with missing data. Statistical methods in medical research, 22(3), 278-295. The phrase IPW should be mentioned in the paper. 3. The missingness is actually stronger than MCAR (missing completely at random). The underlying missingness is actually stronger than MCAR. In equation (3), we see that each coordinate are independent missing. The MCAR allow coordinate to be dependently missing. 4. The problem might be too trivial? There are three components of this papers (that each component can be changed independently). i) regression problem ii) missing data iii) gradient descent In both (i) and (ii), the authors consider the simplest case--linear regression and MCAR+IPW. Both scenarios are well-studied in the literature for decades (including many papers on their combinations). And the MCAR is almost impossible to be true in practice. From a missing data perspective, this problem seem to be too trivial. So it seems to me that the key novelty will be (iii). However, the SGD with averaged iterations seem to be a classical approach in SGD. Thus, I am not sure if the proposed method is novel enough. Note: I am not an expert in optimization so my judge may not be correct. ### comments after reading the rebuttal After reading the authors' rebuttal, I was still not convinced that the missing data part is novel. So I will rely on other reviewer's comment on the contribution of optimization part.


Review 4

Summary and Contributions: Handling missing values with simple imputation (e.g. zero imputation) can add bias to the models. This work proposes a debiased averaged stochastic gradient descent to learn linear regression models (with streaming or finite samples) from data with missing covariates. It can deal with heterogenous missing proportions (e.g. different ratios for different dimensions) and also achieve convergence rate of O(1/n) at iteration n in terms of the generalization risk. This is the same convergence rate for SGD on complete data and gives a state-of-the-art convergence for training debiased linear models with incomplete data.

Strengths: - I appreciate the author’s great effort to proved theoretical proofs with discussions and experimental studies. - The proposed algorithm gives a new state-of-the-art convergence compared to Ma and Needell [15]. - The paper is well-written, and discussions for limitations (e.g. empirical risk) are insightful.
 - The algorithm is neat and practical to train linear models on large scale data. - The paper is relevant to NeurIPS in terms of dealing general problem of training linear scalable models from incomplete data.

Weaknesses: - The algorithm is provided for linear models with convergence guarantees, which is a significant contribution. But, I think it may also practically work for deeper models as well. Maybe a short discussion in appendix (or inside paper) about extension or applications in deeper models may be interesting. - It is just a discussion and not a limitation. I think a parallel direction to this work is how to normalize the inputs to correct the sparsity bias in the model (when doing zero imputation). It has been discussed in the following paper. They propose a sparsity normalization of the input: x = (x \odot mask) / \mu_{mask}, where \mu_{mask} is somehow the same as p in this paper. I think the division by p is reflected in this paper when calculating tilde{g}. So, in overall, there might be some connections between the two directions. Yi, Joonyoung, et al. "Why Not to Use Zero Imputation? Correcting Sparsity Bias in Training Neural Networks."  ** Minor: There are also some typos or incorrect referencing/labeling inside the paper. For example: - The paper starts with theorem 4 instead of 1. - In line 183, we have Remark 3. But, I cannot find - In line 286 and 292, it refers to Table 3. But, it is actually a figure.

Correctness: I cannot verify all the derivations since it is not my exact expertise. But, the paper seems solid.

Clarity: Yes

Relation to Prior Work: Yes. The author may also cite this paper: Yi, Joonyoung, et al. "Why Not to Use Zero Imputation? Correcting Sparsity Bias in Training Neural Networks." 

Reproducibility: Yes

Additional Feedback: ### Post Rebuttal Thanks the authors for their feedback. By referring to Joonyoung et al., I did not want to criticize this submission.I waned to explain the relation. I agree with other reviewers that this algorithm has some limitations such as linearity and MCAR assumption. However, I keep my score as 7 since I think this work has some potentials and contributions such as the the new convergence rate and applications in online ridge regression.

[Author Response · NeurIPS 2020]

We thank all the reviewers for their insightful comments which help us to improve the article. They acknowledged the theoretical work and the relevance of the method developed for the ML community. In the final version, we will integrate the comments of form given by the reviewers (typos, notations, theorem numbering and English writing).

# 1  New experimental results (R2)

**Increasing missing data proportions on synthetic data.**  The figure on the top shows the results of our approach on the same data as in Section 5.1 with 25% (green), 50% (orange) and 75% (red) missing values. It shows that the more missing data there are, the more convergence rate deteriorates. This was expected, as the established theoretical upper bound for the convergence (Th. 4) increases as $p$ gets smaller.

**Comparison to other methods.**  The paper does already include a comparison with the theoretically-grounded competitor for linear regression with missing covariates: the EM algorithm (see Fig. 4), but contrary to our approach, EM requires a distributional assumption on the covariates and prevents from observations in high dimension. Moreover, the proposed strategy results in a practical, efficient and theoretically sound algorithm. For completeness, we ran comparisons on 5 UCI datasets (Boston, Concrete slump, Diabetes, Superconductivity, Wine) to 2-step heuristics: imputation of the covariates (by the mean or ICE[1]) and linear regression (LR) on the completed data, varying the percentage of NA. The coefficient of determination $R^2$ is plotted on the Figure besides (thus higher is better), for the superconductivity dataset, with 60% of missing values. This is representative of other results where our method greatly outperforms the mean imputation and is better or in the same order of magnitude of ICE (that also does not scale well).

# 2  Bibliographic addendum

**IPW. (R3)**  We will include the pointed references on IPW, thank you. While the motivation for reweighting may meet our debiasing will, we would like to point out some differences with our work: in the IPW literature, weighting is often used to rebalance samples with missing outcome, while we consider missing values in all the learning task covariates which can be in high dimension. In addition, the expression we use to debiase SGD, and more specifically its gradients (Eq. 4), although involving weights, is more complex than simply weighting the data.

**Missing values in deep networks. (R4)**  A reference to Joonyoung et al., in which they propose a heuristic to debiase zero-imputation in neural networks, will be added in the final version. Due to the high non-linearity of their setting, their debiasing trick significantly departs from ours, and their proposed algorithm comes with no guarantee of convergence.

# 3  Relevance of the tackled problem and discussion (R1, R3, R4)

**Discussion on the bounded feature assumption (R1)**  As mentioned in the paper, the bounded features assumption is mostly made to ease the readability. It can be actually relaxed into a bound "in average": more precisely only bounds on moments of the random variable can be required, see e.g. Section 6.1. in [1].

**Using estimators of $(p_j)_j$. (R3)**  The available implementation already includes the use of estimated proportions $(\hat{p}_j)_j$ of NA in each column, instead of the oracle ones $(p_j)_j$, and so do all the numerical experiments, always leading to convergent estimators. In addition, we can show that, for the estimator $\hat{\beta}_{k,\hat{p}_j}$ built using our algorithm with $(\hat{p}_j)_j$, *we preserve the optimal $1/k$ convergence rate.* More precisely, the supplementary risk w.r.t. the iterate $\bar{\beta}_k$ built with the true $(p_j)_j$ is $\mathbb{E}[R(\hat{\beta}_{k,\hat{p}_j}) - R(\bar{\beta}_k)] = O(1/kp_{\min}^5)$. A remark and the proof of this preliminary[2] result will be added.

**Towards a more general MCAR setting. (R3)**  We indeed considered a specific MCAR setting, in which the missing-pattern random variables were independent. We thank R3 from raising this interesting issue: we can extend the setting to allow coordinates to be dependently missing. To do so, we propose a new way of constructing debiased versions of gradients (Lemma 1), as $\tilde{g}_k(\beta) := (W \odot (\tilde{X}_{k:}\tilde{X}_{k:}^T))\beta - y_k P^{-1}\tilde{X}_{k:}$ with $W \in \mathbb{R}^{d \times d}$, and $W_{ij} := 1/\mathbb{E}[\delta_{ki}\delta_{kj}]$. The noise structure in the SGD iterates (Lemma 2(3.)) becomes even more technical to control. Regarding practical implementation, the matrix $W$ can be estimated, in particular using low-rank strategies on the missing pattern matrix.

**Impact of our work and extensions. (R3,R4)**  Despite the apparent simplicity of the considered setting, we tackle the important issue of performing large-scale Ridge regression with missing covariates (which was not resolved yet), using SGD, thereby handling large dimensionality and online (missing) data. This actually required a lot of technicality: the state-of-the-art proofs cannot be applied directly. An efficient code is also provided. SGD being a keystone block of ML, this paves the way of many developments to handle ubiquitous missing data (e.g., variance reduction, deep learning, etc.). Moreover, theoretical locks are raised, cleared up or resolved: (i) theoretical challenge of multi-pass ERM, (ii) computational optimality with missing data (Th. 4), (iii) information theory optimality (see Sec. 4.4, with an open question to establish a lower bound).

[1] Dieuleveut, Durmus, and Bach. Bridging the gap between constant step size SGD and markov chains. *Ann. Statist.*, 48, 2020.

## Footnotes

[1] `sklearn.impute.IterativeImputer`   [2] the dependency in $p_{\min}$ may not be optimal and the proof requires the invertibility of the covariates' covariance matrix $H$ and bounded iterates.


[Meta-Review · NeurIPS 2020]

The paper addresses the issue of missing values in SGD. There was agreement that the obtained results are novel but there was no strong advocate for the paper. The general consensus is that the paper is borderline